# Lossless Compression: A New Benchmark for Time Series Model Evaluation

## Abstract

The evaluation of time series models has traditionally focused on four canonical tasks: forecasting, imputation, anomaly detection, and classification. Although these tasks have made significant progress, they primarily assess task-specific performance and do not rigorously measure whether a model captures the full generative distribution of the data. We introduce lossless compression as a new paradigm for evaluating time series models, grounded in Shannon's source coding theorem. This perspective establishes a direct equivalence between optimal compression length and the negative log-likelihood, providing a strict and unified information-theoretic criterion for modeling capacity. Then we define a standardized evaluation protocol and metrics. We further propose and open-source a comprehensive evaluation framework TSCom-Bench, which enables the rapid adaptation of time series models as backbones for lossless compression. Experiments across diverse datasets on state-of-the-art models, including TimeXer, iTransformer, and PatchTST, demonstrate that compression reveals distributional weaknesses overlooked by classic benchmarks. These findings position lossless compression as a principled task that complements and extends existing evaluations for time series modeling.

## 1 Introduction

Time series modeling is a fundamental branch of machine learning with critical applications in finance, healthcare, climate science, and industrial operations Sakib et al. (2025). Recent advances in deep learning have pushed the field from early recurrent and convolutional networks to models utilizing self-attention and hybrid architectures, which demonstrate remarkable performance across a variety of settings Kim et al. (2025); Mahmoud & Mohammed (2024). However, a central challenge remains unresolved: how to systematically and rigorously evaluate their modeling capacity.

Currently, the time series research widely relies on four canonical benchmark tasks: forecasting, anomaly detection, imputation, and classification Jin et al. (2024). While these tasks have undeniably advanced the field, they exhibit an inherent limitation: their optimization objectives do not directly correspond to a model's ability to capture the global statistical structure of a sequence. In other words, they primarily validate task-specific functionality but fail to provide a comprehensive assessment of distributional modeling capacity. Specifically, forecasting tasks typically minimize MSE or MAE, which can be satisfied by short-term lags or average baselines while overlooking tail risks and regime shifts Jean (2025). Classification tasks may achieve high accuracy by focusing on a few features strongly correlated with labels, ignoring the majority of temporal dependencies Sun et al. (2024). Imputation tasks are optimized under artificially masked conditions, emphasizing local consistency rather than global distributional fidelity Zhang et al. (2024). Anomaly detection emphasizes distinguishing between "normal" and "abnormal" boundaries Lee et al. (2024). Therefore, these four tasks are closer to functional validation. They can demonstrate that a model is useful in specific applications, but they cannot answer a deeper question: does the model truly capture the entropy structure and generative regularities of time series?

Addressing this gap requires an evaluation perspective that directly characterizes the generative distribution rather than merely assessing task-specific performance. Lossless compression in information theory provides precisely such a bridge. Recent studies have highlighted a close connection between language modeling and lossless compression. DeepMind's work formalizes that autore-

gressive models paired with arithmetic coding act as universal compressors Delétang et al. (2023a). Marcus Hutter, founder of the *Hutter Prize*, argues that intelligence can be measured by the ability to compress data effectively Kipper (2021). For time series, the connection with lossless compression is even more natural Wan et al., as the act of predicting each subsequent byte is a granular test of the model's ability to approximate the true conditional probability of the underlying data-generating process Mao et al. (2022). A model that achieves strong compression must have learned to represent complex, multi-level dependencies in a compact, low-entropy form Delétang et al. (2023a). Furthermore, much like forecasting or classification which are valuable applications, lossless compression is a critical real-world task for efficient data storage and transmission Elakkiya & Thivya (2022). Therefore, our work innovatively introduces lossless compression as a new benchmark for time series evaluation. The main contributions of this work are summarized as follows:

- **A novel evaluation task:** We introduce lossless compression as an independent benchmark task, complementing and extending the existing four canonical tasks.
- **Theoretical grounding:** We rigorously derive the equivalence between compression objectives and probabilistic modeling goals, highlighting its unique role in optimization, information constraints, and modeling granularity.
- **Pluggable compression framework:** We propose and open-source *TSCom-Bench*, a standardized lossless compression evaluation framework that allows seamless integration of time series models as backbones and outputs a comprehensive suite of evaluation metrics.
- **Comprehensive empirical study:** We conduct extensive experiments on diverse real-world and synthetic datasets, benchmarking both classical compressors and modern learning-based time series models.

## 2 PRELIMINARIES AND MOTIVATION

### 2.1 MULTIVARIATE TIME SERIES AND OPTIMAL CODE LENGTH

We consider a multivariate time series $X = \{x_t \in \mathbb{R}^d\}_{t=1}^T$, where $T$ is the total time steps and each observation $x_t \in \mathbb{R}^d$ is a $d$-dimensional vector at a given time step $t$, with $d$ denoting the number of channels. From an information-theoretic perspective, the $x_{<t} = (x_1, \ldots, x_{t-1})$ denotes the history of observations before $t$, the goal is equivalent to accurately approximating the true conditional probability. According to Shannon's source coding theorem Barron et al. (1998), the theoretical optimal expected code length of $X$ under an ideal entropy coder is asymptotically equal to its negative log-likelihood (NLL):

$$L^*(X) = -\sum_{t=1}^T \log_2 P(x_t \mid x_{<t}), \tag{1}$$

where $L^*(X)$ is the optimal code length in bits required to encode the entire sequence $X$. The term $P(x_t \mid x_{<t})$ within the summation is the true conditional probability of observing $x_t$ given all previous observations $x_{<t}$. This equivalence implies that a model's ability to compress a time series is a direct measure of how well it approximates the true data-generating process Gruver et al. (2023).

### 2.2 FROM MULTIVARIATE TIME SERIES TO SYMBOLIC STREAMS

To apply compression-based evaluation, the continuous time series $X$ must be mapped to a discrete sequence. Let $f : \mathbb{R}^d \to \mathcal{A}^k$ be a bijective encoding function, where $\mathcal{A}$ is a finite alphabet (e.g., bytes, where $|\mathcal{A}| = 256$) and $k$ is the number of symbols required to represent a single real number (e.g., $k = 4$ for a 32-bit float). Assuming a homogeneous data type across all channels. This function maps the time series $X$ to a symbolic stream $S$:

$$S = f(X) \in \mathcal{A}^L, \quad \text{where } L = T \cdot d \cdot k. \tag{2}$$

Here, $S$ is the resulting byte stream, and $L$ is the total length in bytes. If the encoding function $f$ is bijective, then the Shannon entropy measured in bits, using base-2 logarithms $\log_2$, denoted by $H(\cdot)$, is preserved between the original time series $X$ and its encoded stream $S$:

$$H(X) = H(S). \tag{3}$$

This equality holds exactly under a perfect bijective mapping. In practice, when continuous values are quantized, a small approximation error may occur, but it vanishes as the quantization becomes infinitely precise (Cover & Thomas, 2006). Therefore, byte-level compression faithfully reflects the probabilistic modeling quality for real-valued multivariate time series.

### 2.3 COMPRESSION OBJECTIVE AND KL DIVERGENCE

The central quantity in compression is the expected code length. For a byte stream $S$ drawn from the true data distribution $P$, a model $Q_\theta$ parameterized by $\theta$ assigns a likelihood via an autoregressive factorization:

$$Q_\theta(S) = \prod_{i=1}^{L} Q_\theta(s_i \mid s_{<i}), \tag{4}$$

where $s_i$ is the $i$-th symbol in the stream $S$ of total length $L$, and $s_{<i}$ denotes the history of preceding symbols. The compression loss $\mathcal{L}_{\text{comp}}$ is defined as the expected negative log-likelihood:

$$\mathcal{L}_{\text{comp}}(\theta) = \mathbb{E}_{S \sim P}\Big[ -\log_2 Q_\theta(S)\Big]. \tag{5}$$

This loss decomposes into Shannon entropy and KL divergence:

$$\mathcal{L}_{\text{comp}}(\theta) = H(P) + \text{KL}(P\|Q_\theta), \tag{6}$$

where $H(P)$ is the Shannon entropy of the true distribution $P$, and $\text{KL}(P\|Q_\theta)$ is the Kullback-Leibler (KL) divergence between $P$ and $Q_\theta$. Thus, minimizing $\mathcal{L}_{\text{comp}}$ is equivalent to minimizing the KL divergence, which forces the model distribution to align with the true data distribution. The derivation process establishes compression as the most principled evaluation: only if a model fully captures the distribution will it achieve near-optimal compression.

## 3 OVERALL COMPRESSION ARCHITECTURE

The overall lossless compression evaluation architecture integrates byte stream serialization, time series probabilistic modeling, and arithmetic encoding into a unified pipeline, as shown in Figure 1. First, the uncompressed file is read as a byte stream, forming the byte stream serialization $(s_1, s_2, \ldots, s_{i-1})$ that is fed into the time series model to derive the probability distribution $Q_\theta$ of the next byte $s_i$. Then, these probability vectors are fed into an arithmetic encoder for arithmetic encoding. The arithmetic encoder is a standard entropy coding algorithm that first performs cumulative probability calculation, then iteratively reduces the unit interval based on the predicted probabilities to assign each byte to a sub-interval. Through continuous interval narrowing, the entire sequence is represented by a final interval. This final interval is converted into the shortest binary fraction to generate a compressed bitstream that ultimately forms the compressed file. This compressed file can be accurately decoded back to the original file through reverse processing. Thus, this architecture unifies probabilistic modeling and compression, which is reflected in the fact that the more accurately a time series model captures temporal dependencies, the more efficient its compression becomes.

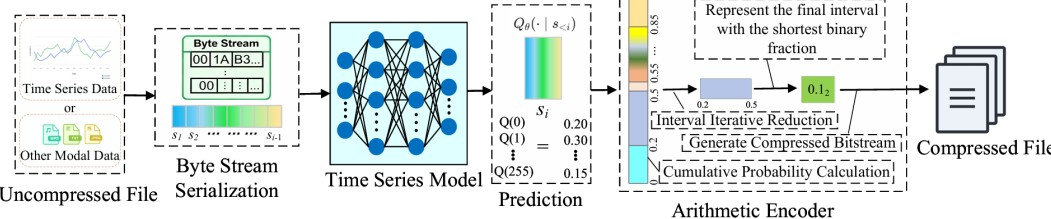

Figure 1: Overall lossless compression architecture. Byte-level encoding, probabilistic modeling, and arithmetic coding are combined into a unified pipeline.

# 4 COMPARISON WITH CANONICAL TASKS

We provide a comparison between lossless compression and the four canonical evaluation tasks widely used in time series modeling: forecasting, imputation, anomaly detection, and classification. The differences in evaluation of these tasks will be discussed in the appendix.

**Unified View.** The canonical tasks can be abstractly interpreted as minimizing a divergence between projected statistics of the true and model distributions. This can be conceptualized as:

$$\mathcal{L}_{\text{task}}(\theta) \approx d(\phi(P), \phi(Q_\theta)), \tag{7}$$

where $\mathcal{L}_{\text{task}}$ represents a generic task loss, $\phi$ is a function that extracts a relevant statistic (e.g., the conditional mean for forecasting), and $d(\cdot, \cdot)$ is a generic distance or divergence measure. These projections constrain only partial aspects of the distribution.

**Illustrative Counterexample.** Consider a time series generated by a binary mixture process. For any history $x_{<t}$, the next value $x_t$ is drawn from the conditional distribution:

$$P(x_t \mid x_{<t}) = \frac{1}{2}\delta(x_t - (\mu - a)) + \frac{1}{2}\delta(x_t - (\mu + a)), \tag{8}$$

where $\mu, a \in \mathbb{R}$ with $a > 0$ are fixed constants, and $\delta(\cdot)$ is the Dirac delta function, which we use to compactly represent a two-point discrete distribution. The conditional mean of this process is always $\mathbb{E}_p[x_t \mid x_{<t}] = \mu$. A forecasting model that always predicts this conditional mean, $\hat{x}_t = \mu$, achieves an MSE of:

$$\mathbb{E}_p\big[(x_t - \mu)^2\big] = a^2, \tag{9}$$

which is the optimal solution for minimizing MSE. For a conceptual illustration, suppose a model $Q_\theta$ incorrectly assumes a narrow Gaussian distribution, $\mathcal{N}(\mu, \sigma^2)$, where the variance $\sigma^2 \ll a^2$. This model's mean prediction is also $\mu$, so its MSE remains near-optimal. However, its compression performance, measured by the cross-entropy $-\log_2 Q_\theta(x_t \mid x_{<t})$ will be extremely poor. The model $Q_\theta$ assigns negligible probability density to the only two points that can actually occur, $x_t = \mu \pm a$, causing the negative log-likelihood to diverge towards infinity. Therefore, a model can appear successful under forecasting metrics while failing under compression, which demonstrates that compression provides a stricter and more informative evaluation.

# 5 BENCHMARK DESIGN AND METHODOLOGY

We propose a standardized benchmark that evaluates time series models via lossless compression, providing a rigorous and reproducible methodology and protocols.

## 5.1 ENCODING CONVENTIONS

To guarantee both losslessness and reproducibility, we recommend a canonical encoding scheme:

- **Numeric representation.** Each real-valued observation is stored in IEEE-754 32-bit/UTF-8 format (16/64-bit can be evaluated in ablations). Every float is decomposed into $k = 4$ bytes, each a symbol from $\mathcal{A}$ with $|\mathcal{A}| = 256$. Bytes are concatenated in a fixed order (channel-first, then time), yielding the symbol stream $S = f(X)$.
- **Bijectivity.** The mapping $f : X \mapsto S$ is deterministic and invertible, ensuring exact recovery of the original sequence via $f^{-1}$.
- **Preprocessing.** Any preprocessing (e.g., missing value imputation, normalization, boundary alignment) must be standardized and released with the dataset package.
- **Alternative encodings.** Other discretization schemes (e.g., histogram binning, lossy quantization) may be studied, but benchmark results should always report the canonical byte-level encoding for comparability.

## 5.2 MODEL-TO-CODER INTERFACE

Time series models are treated as *predictors* that interface with a lossless entropy coder.

- **Interface.** For each prefix $s_{<i}$, the model outputs a probability vector $Q_\theta(\cdot \mid s_{<i})$ over $\mathcal{A}$.
- **Training paradigms.** Two primary training paradigms are supported: (i) Autoregressive models are trained directly on symbol streams (default); or (ii) density estimators are trained on raw values and subsequently mapped to discrete probabilities.
- **Entropy coder.** An arithmetic coder consumes the probability vectors together with the ground-truth sequence $S$. Encoding length equals the negative log-likelihood.
- **Numerical stability.** Probability vectors must be properly normalized; log-space accumulations or fixed-precision mappings are recommended to avoid underflow or mismatch.

## 5.3 EVALUATION PROTOCOL AND METRICS

To ensure comparability, models are trained on the designated training split and evaluated on held-out test sequences, with no adaptive coding across training and test allowed. All preprocessing, random seeds, and hyperparameters should be fixed and released to ensure strict reproducibility. We report metrics for both compression efficiency and runtime. These include bits per byte (bpb), compression ratio (CR), and Compression Throughput (CT), defined as:

$$\text{bpb} = \frac{L_{\text{comp}}(Q_\theta, S)}{L}, \quad \text{CR} = \frac{L_{\text{comp}}(Q_\theta, S)}{8 \cdot L}, \quad \text{CT} = \frac{L/1024}{T_{\text{compress}}}, \tag{10}$$

where $L_{\text{comp}}(Q_\theta, S)$ is the total compressed length in bits, $L$ is the original length of the byte stream $S$ in bytes, and $T_{\text{compress}}$ is the compression time in seconds.

## 5.4 OPEN-SOURCE TSCOM-BENCH FRAMEWORK

Models in TSCom-Bench are evaluated in their standard architectural form. We do not change the backbone structure. It is worth noting that we are a new compression task parallel to prediction, classification tasks, etc., and will not perform secondary fine-tuning based on the training model. Any autoregressive backbone used for forecasting or classification can be adapted with very little code, usually fewer than 20 lines of code. We strongly encourage releasing preprocessing code, training scripts, and entropy coding implementations. All components of this benchmark have been open-sourced in the **TSCom-Bench** framework, which provides standardized encoding functions, reference coders, datasets, and evaluation scripts for direct and reproducible comparison. Codes are available in `https://anonymous.4open.science/r/TSCom-Bench-8262`.

# 6 EXPERIMENTS

## 6.1 EXPERIMENTAL SETUP

**Datasets.** We evaluate on a diverse collection of widely used multivariate time series benchmarks, including PEMS08, Traffic, Electricity, Weather, ETTh2 and Solar datasets. For PEMS08 we follow standard practice and use the publicly released compressed NumPy archive (`.npz`), whose byte stream is already stored in a ZIP-based container and later serves as a negative control for calibrating our benchmark. In addition, we include standard lossless compression benchmarks such as Enwik9 (Wikipedia text), Image (raw image bitmaps), Sound (audio waveforms), Float, Silesia and Backup archives.

**Baselines.** We compare against representative state-of-the-art forecasting backbones widely adopted in time series research, including Transformer-based models Informer Zhou et al. (2021), Autoformer Wu et al. (2021), PatchTST Nie et al. (2022), SCINet Liu et al. (2022), iTransformer Liu et al. (2023), TimeXer Wang et al. (2024), lightweight linear approaches DLinear Zeng et al. (2023) and recent hybrid architectures LightTS Campos et al. (2023). Classical compressors such as Dzip Goyal et al. (2021) and NNCP Bellard (2019) is also included for reference.

**Environments and Parameters.** All experiments are implemented in PyTorch 2.1 and executed on NVIDIA Tesla P100 GPUs. For neural baselines, we adopt standard training protocols following prior work: the sequence length is fixed at 96, and data are normalized with RevIN preprocessing. Optimization uses Adam with learning rates selected from $\{10^{-3}, 10^{-4}\}$, and employs early stopping based on validation loss. For evaluation, we report bpb, CR and CT for comparison.

Table 1: Lossless compression results on six benchmark time series datasets. CT is measured in KB/s. The best results are highlighted in **bold**, and the second best are underlined.

| Dataset | TimeXer (2025) | | iTransformer (2024) | | PatchTST (2023) | | Autoformer (2023) | | DLinear (2023) | | LightTS (2023) | | SCINet (2022) | | Informer (2021) | |
|---|---|---|---|---|---|---|---|---|---|---|---|---|---|---|---|---|
| | CR | CT | CR | CT | CR | CT | CR | CT | CR | CT | CR | CT | CR | CT | CR | CT |
| PEMS08 | **0.978** | 12.55 | 0.978 | 18.13 | 0.978 | 9.63 | 0.980 | 3.24 | 0.996 | **30.92** | 0.989 | 17.41 | 0.980 | 2.74 | 0.979 | 2.74 |
| Traffic | **0.137** | 15.58 | 0.141 | 23.21 | 0.137 | 11.89 | 0.151 | 3.27 | 0.155 | **60.06** | 0.174 | 24.63 | 0.140 | 1.29 | 0.167 | 4.18 |
| Electricity | **0.112** | 16.19 | 0.142 | 23.06 | 0.115 | 12.32 | 0.194 | 3.26 | 0.176 | **57.79** | 0.168 | 24.33 | 0.135 | 2.87 | 0.194 | 4.17 |
| Weather | **0.207** | 15.63 | 0.268 | 21.99 | 0.213 | 11.76 | 0.370 | 2.15 | 0.382 | **54.57** | 0.370 | 20.56 | 0.332 | 3.52 | 0.418 | 2.77 |
| ETTh2 | **0.262** | 15.04 | 0.364 | 20.50 | 0.285 | 11.67 | 0.404 | 2.17 | 0.495 | **44.72** | 0.534 | 22.13 | 0.412 | 3.53 | 0.437 | 2.74 |
| Solar | **0.027** | 16.61 | 0.036 | 24.70 | 0.029 | 21.98 | 0.074 | 2.79 | 0.068 | **65.55** | 0.055 | 27.22 | 0.049 | 2.90 | 0.093 | 2.79 |

Table 2: CR under the MSCI setting on four multivariate time series datasets.

| Dataset | iTransformer | TimeXer | PatchTST | SCINet | Informer | Autoformer | DLinear | LightTS |
|---|---|---|---|---|---|---|---|---|
| Weather | **0.1581** | 0.1651 | 0.1690 | 0.2664 | 0.2727 | 0.3078 | 0.4545 | 0.3485 |
| ETTh2 | 0.2127 | **0.2106** | 0.2160 | 0.2203 | **0.2106** | 0.2185 | 0.2845 | 0.3121 |
| Electricity | 0.0816 | **0.0787** | 0.0808 | 0.0873 | 0.0862 | 0.0916 | 0.1939 | 0.1487 |
| Traffic | 0.1807 | 0.1076 | **0.1068** | 0.1251 | 0.1293 | 0.1295 | 0.2491 | 0.2501 |

## 6.2 MAIN RESULTS: LOSSLESS COMPRESSION ACROSS TIME SERIES BENCHMARKS

To validate lossless compression as a principled evaluation paradigm for time series modeling, we conduct systematic experiments across six real-world benchmark datasets, with results summarized in Table 1. Two points are worth highlighting. The Solar's remarkably low CR directly reflects its minimal data entropy, which stems from a highly predictable diurnal cycle and inherent sparsity from frequent zero-values during nighttime. This ability to quantify the data's intrinsic predictability is a crucial insight inaccessible to classic error-based metrics. In contrast, PEMS08 consistently shows CR values close to 1, consistent with the results for general-purpose compressors in Appendix Table 10, indicating near-incompressibility. The fact that our pipeline correctly identifies this pre-compressed data as having minimal remaining redundancy serves as a crucial validation of its correctness and reliability.

The results across all datasets reveal that leading models like TimeXer, iTransformer and PatchTST consistently demonstrate strong performance on the compression task, aligning with their effectiveness in other tasks. An interesting finding is that PatchTST's superior compression, despite not always leading in forecasting, indicates its ability to capture rich distributional representations overlooked by task-specific objectives. Overall, these results demonstrate that lossless compression provides a more fundamental and stringent benchmark, exposing differences and limitations invisible to functional evaluations and supporting its role as a core benchmark for time series models.

## 6.3 MULTI-STREAM CHANNEL-INDEPENDENT (MSCI) SETTING

To fully exploit models such as TimeXer, iTransformer and PatchTST that contain channel-aware components, we conduct a multi-stream version of the experiment. Specifically, we treat each variable in the dataset as an independent read channel. A single model instance processes and compresses each channel in sequence, and the final file size is obtained by summing over all channels. Across all datasets, the MSCI setting yields lower CR than the single-stream setting (see Table 2). This indicates that, when channel boundaries are preserved, multivariate data contains structural information beyond temporal continuity, and this structure becomes clearer and easier to learn. Channel-independent models, especially iTransformer and TimeXer, benefit the most, confirming that their CI design indeed captures meaningful per-channel temporal patterns.

Table 3: Results of time series and IEEE 754 structure ablation CR experiments on WEATHER dataset. "A" retains both time series and IEEE 754 structure; "B" removes only the time series; "C" removes both.

| Method | A Raw | B Shuffled time | C Shuffled time and bytes |
|---|---|---|---|
| TimeXer | 0.3434 | 0.4909 | 0.7690 |
| iTransformer | 0.3718 | 0.5106 | 0.7691 |
| PatchTST | 0.3485 | 0.4849 | 0.7705 |
| LightTS | 0.6654 | 0.7447 | 0.8243 |

## 6.4 LEARNING TEMPORAL DEPENDENCIES IN BYTE-LEVEL ENCODING

A natural concern for our byte-level framework is that splitting a 32-bit IEEE 754 float into four bytes might destroy useful structure: it could weaken temporal dynamics in the original sequence and break the internal sign–exponent–mantissa dependency. To verify that these potential information structures can actually be learned by the model, we design the following controlled experiment.

We use the Weather datasets and consider three settings:

- **A: Raw data.** The original time series is encoded into bytes in temporal order. Both temporal structure and IEEE 754 structure are preserved.

- **B: Shuffled time.** We randomly permute the time steps before encoding. Temporal order is removed, while the IEEE 754 layout within each value is preserved.

- **C: Shuffled time and shuffled bytes.** We randomly permute both the time steps and the four bytes inside each 32-bit float. Both temporal and IEEE 754 structures are removed.

In all three settings we keep the same models, training protocol, and compression metric. Table 3 shows the results. From setting A to B, the CR increases consistently across all models, even though the IEEE 754 structure inside each float remains unchanged. This indicates that models rely on temporal dynamics such as trend and seasonality. If byte-level encoding had destroyed temporal information, shuffling the time index would not cause such a clear and systematic drop in compression performance.

From setting B to C, the metric becomes even worse. The only additional change is shuffling the four bytes within each float, which breaks the deterministic relation between sign, exponent, and mantissa. The consistent degradation from B to C suggests that models also learn this internal numeric structure: they capture dependencies between the first byte (sign and exponent) and the subsequent bytes that refine the mantissa.

Overall, these results demonstrate that byte-level lossless compression preserves both macro temporal structure and micro numeric structure. Models can still learn temporal dependencies across time steps while also capturing the internal IEEE 754 layout within each value, even though the data is presented as a flat byte stream.

## 6.5 CONVERGENCE TO THE ENTROPY LIMIT ON SYNTHETIC DATA

To directly assess whether our approach can recover the true underlying data-generating distribution rather than overfitting to local repetitions, we construct a controlled synthetic dataset. This dataset consists of discrete-valued samples generated with a fixed period of 1,000 bytes and small additive noise, producing an approximately Gaussian marginal value distribution with nontrivial temporal regularity. Figure 2 shows two aspects of this experiment. Panel (a) illustrates a segment of the periodic byte sequence, where the repeated structure and injected noise are clearly visible. Panel (b) compares the original and model-predicted byte-level distribution trends: the strong overlap between the red and green curves indicates that the model successfully captures the global statistical properties of the data rather than merely memorizing individual cycles or local patterns. We then evaluate the learned model using our lossless compression protocol. As shown in Table 4, the theoretical lower bound of the compression rate is approximately 1.0097 bpb, with small fluctuations

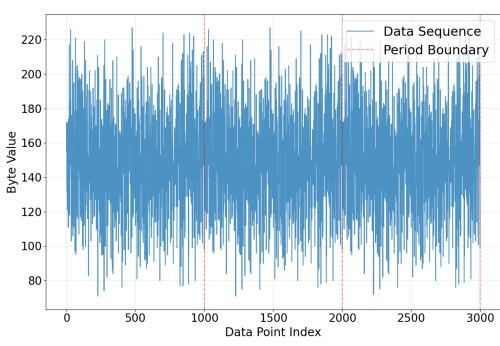

(a) Segment of the periodic synthetic byte sequence.

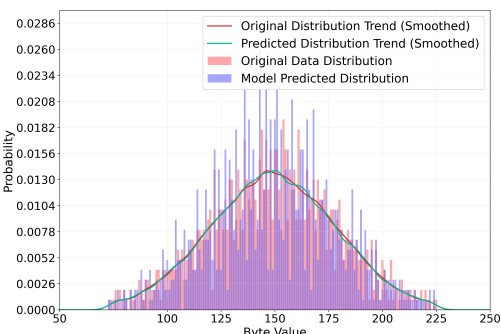

(b) True vs. model-predicted byte-level distributions.

Figure 2: Synthetic data entropy validation.

due to injected noise. As the dataset size increases, the gap between the model's bpb and the bound steadily decreases, demonstrating clear convergence toward the information-theoretic limit.

This experiment provides two key insights for our benchmark. First, it confirms that lossless compression evaluation reflects a model's ability to recover global statistical regularities. Second, it shows that as more data is observed, a well-specified model can approach the entropy limit, which serves as a rigorous, interpretable upper bound for modeling capacity.

Table 4: Empirical compression converges to theoretical entropy on synthetic data.

| Metric | 1MB | 2MB | 4MB | 8MB | 16MB | 32MB | 128MB |
|---|---|---|---|---|---|---|---|
| True Entropy | 1.0087 | 1.0066 | 1.0089 | 1.0097 | 1.0090 | 1.0097 | 1.0097 |
| Model bpb | 1.1251 | 1.0945 | 1.0639 | 1.0482 | 1.0347 | 1.0301 | 1.0442 |
| Gap | 0.1154 | 0.0848 | 0.0542 | 0.0385 | 0.0250 | 0.0204 | 0.0345 |

### 6.6 CROSS-MODALITY COMPRESSION BENCHMARK

To evaluate whether lossless compression truly captures cross-domain temporal regularities, we further construct a multimodal compression benchmark by interleaving heterogeneous data audio segments, environmental sensor readings, and textual event into a unified IEEE-754/UTF-8 byte stream following our canonical encoding. This setting mimics real-world archives where diverse modalities must be stored jointly without loss. As shown in Table 5, time-series models consistently outperform classical compressors such as Dzip and NNCP even under cross-modal interleaving, with TimeXer achieving the lowest CR of 0.185 while maintaining high CT on Enwik9. These results provide direct evidence that temporal modeling for compression generalizes beyond single-modality data and yields superior compression efficiency on heterogeneous multimodal streams. The results highlight that incorporating compression as a task is not only a theoretical exercise for model evaluation, but also directly addresses the practical need for efficient data archival in real-world applications.

### 6.7 RELATIONSHIP BETWEEN COMPRESSION AND CLASSIC TIME SERIES TASKS

To investigate how lossless compression relates to classic time series tasks, we compare our compression evaluations with publicly reported results on forecasting, imputation, anomaly detection, and classification. The results for representative models are collected from their original benchmark papers and widely used survey tables Wang et al. (2024); Liu et al. (2023); Wu et al. (2022). Lossless compression results are taken from our standardized TSCom-Bench evaluation protocol in Table 1. For comparability across heterogeneous metrics, all task scores are normalized to the range $[0, 1]$ within each task. The radar plot in Figure 3 (a) displays the normalized scores across five tasks, revealing distinctive performance profiles: models such as iTransformer achieve strong forecasting and imputation results but lag markedly on compression, forming an asymmetric profile. In contrast,

Table 5: Lossless compression results on seven compression-benchmark cross-modality datasets. The best results are highlighted in **bold**, and the second best are underlined.

| Dataset | TimeXer (2025) | | iTransformer (2024) | | PatchTST (2023) | | DLinear (2023) | | SCINet (2022) | | Dzip (2021) | | NNCP (2019) | |
|---|---|---|---|---|---|---|---|---|---|---|---|---|---|---|
| | CR | CT | CR | CT | CR | CT | CR | CT | CR | CT | CR | CT | CR | CT |
| Enwik9 | **0.185** | 14.35 | 0.206 | 16.67 | 0.187 | 13.21 | 0.359 | **32.54** | 0.263 | 3.64 | 0.224 | 4.06 | 0.279 | 1.05 |
| Sound | **0.431** | 13.67 | 0.479 | 25.54 | 0.455 | 10.37 | 0.592 | **40.63** | 0.535 | 1.69 | 0.490 | 4.51 | 0.615 | 1.13 |
| Image | **0.517** | 18.43 | 0.615 | 24.57 | 0.523 | 14.12 | 0.741 | **38.42** | 0.713 | 2.95 | 0.581 | 4.77 | 0.676 | 1.32 |
| Float | 0.312 | 14.53 | 0.327 | 19.56 | **0.291** | 12.35 | 0.392 | **53.67** | 0.429 | 1.72 | 0.694 | 4.51 | 0.582 | 1.23 |
| Silesia | **0.198** | 17.04 | 0.202 | 23.64 | 0.207 | 13.74 | 0.425 | **48.96** | 0.402 | 2.82 | 0.209 | 4.79 | 0.395 | 1.26 |
| Backup | **0.528** | 17.25 | 0.575 | **22.83** | 0.552 | 22.34 | 0.730 | 39.78 | 0.647 | 1.96 | 0.572 | 5.11 | 0.598 | 1.65 |

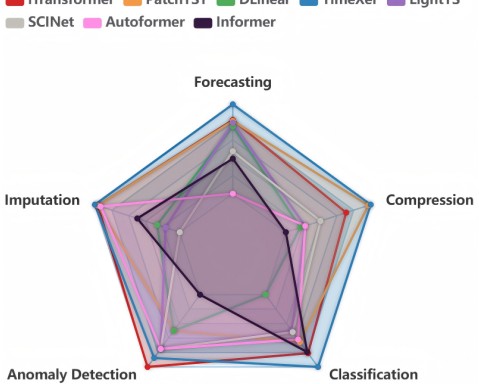

(a) Normalized performance across five tasks.

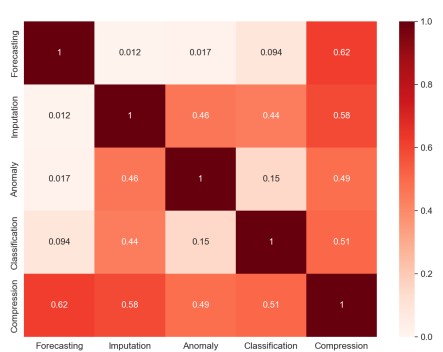

(b) Pairwise correlations between tasks.

Figure 3: Relationship between compression and classic time series tasks from publicly reported benchmarks. (a) Radar plot compares representative models on forecasting, imputation, anomaly detection, classification, and compression tasks. (b) Correlation matrix quantifies task relationships. Compression scores are from our lossless evaluation on the Weather dataset.

TimeXer and PatchTST maintain relatively balanced performance across all dimensions. Figure 3 (b) quantifies these relationships via the Pearson correlation between normalized task performances. The four classic tasks show no consistent or universal correlation pattern with each other, reflecting their focus on different aspects of time series behavior. In contrast, lossless compression exhibits a moderate and relatively uniform correlation with all these tasks. This pattern suggests that compression reflects a model's ability to approximate the global data distribution rather than being tied to any single local objective.

This observation points to a promising direction: training models with compression-oriented objectives could provide a strong pretraining backbone, with task-specific heads fine-tuned for forecasting, imputation, anomaly detection, or classification. Such a framework may unify evaluation and pretraining for time series modeling, analogous to language modeling in NLP. Details of the task metrics, normalization, and data sources are provided in the Appendix for reproducibility.

## 7 RELATED WORK

### 7.1 LOSSLESS COMPRESSION AND INFORMATION-THEORETIC EVALUATION

Shannon's source coding theorem and the close relation between negative log-likelihood and optimal code length form the theoretical backbone connecting probabilistic modeling and compression Cover & Thomas (2006). The use of compression as a measure of model quality has a long history in algorithmic information theory and minimum description length (MDL) principles Rissanen

(1978); Grünwald (2007). Hutter and colleagues formalized connections between induction, intelligence and compression in the context of Solomonoff induction and universal prediction Hutter (2005). Recent work in the deep learning era has revisited compression as a principled evaluation approach for language models and generative systems Delétang et al. (2023b); Yang et al. (2025). Our work adapts these information-theoretic perspectives specifically to multivariate time series, providing practical encoding and evaluation protocols targeted at modern time series architectures.

## 7.2 LEARNING-BASED COMPRESSION AND PROBABILISTIC SEQUENCE MODELING

Traditional lossless compressors such as LZ-family, gzip and bzip2 rely on dictionary or statistical coding heuristics and are effective for certain data modalities Ziv & Lempel (1977). Neural and learning-based compressors employ learned probability models (autoregressive models, VAEs with entropy models, flow-based models) together with arithmetic/ANS coders to achieve superior compression for images, audio and text Ballé et al. (2017); van den Oord et al. (2016); Sain et al. (2023). In the sequence domain, autoregressive models (RNNs, Transformers) serve as learned predictors to drive entropy coding; notable examples include language modeling-based compressors and recent transformer-based compression efforts Rae et al. (2020); Bellard (2020). For time series specifically, prior work has considered both lossy and lossless approaches, including predictive coding, differencing and domain-specific encoders Chiarot & Silvestri (2022). The recent SEP framework improves the speed and memory efficiency of existing models through GPU-level optimizations, while a semantic enhancement module boosts the compression ratio Wan et al.. However, a systematic benchmark that treats lossless compression itself as a canonical evaluation task for general-purpose time series models has not been established. TSCom-Bench seeks to fill this gap by formalizing encoding conventions, evaluation metrics and baselines compatible with contemporary time series architectures such as iTransformer and TimeXer Liu et al. (2023); Wang et al. (2024).

## 7.3 LOSS–METRIC MISMATCH

The mismatch between optimization objectives and evaluation metrics is a well established topic in machine learning, and our empirical finding in time series is a concrete instance of this broader phenomenon. Specifically, Theis et al. (2015) provide a theoretical justification showing that likelihood and sample quality do not necessarily correlate, highlighting that the training objective may not reflect true model performance. Elmachtoub & Grigas (2022) demonstrate that minimizing mean squared error in forecasting does not ensure optimal downstream utility in real decision settings, indicating that MSE often functions only as a surrogate objective. Stein et al. (2023) further show that modern generative modeling metrics may not faithfully capture actual modeling quality.

This work provides an empirical verification of this phenomenon in the time series domain. Many SOTA forecasting models achieve competitive MSE performance yet perform significantly worse under lossless compression, which corresponds to evaluating negative log-likelihood, and lossless compression thereby provides a unified information-theoretic view for revealing this form of metric mismatch.

## 7.4 CONCLUSION

In this paper, we propose lossless compression as a new benchmark for evaluating time series models and release the open-source TSCom-Bench framework to standardize its evaluation. Our experiments demonstrate that this information-theoretic metric reveals distributional weaknesses in SOTA models that are overlooked by conventional tasks. We advocate for its adoption as a new canonical benchmark, as it not only provides a more stringent evaluation of models but also constitutes an indispensable real-world application. Looking forward, we believe this approach offers a powerful pre-training strategy, where models pre-trained on the compression objective can then be fine-tuned for downstream tasks such as forecasting or classification.

## ETHICS STATEMENT

This research focuses on foundational methods using public, anonymized datasets and does not present any foreseeable ethical concerns or negative societal impacts.

REPRODUCIBILITY STATEMENT

We are committed to ensuring the full reproducibility of our research. The source code for our proposed TSCom-Bench framework, which includes implementations of the evaluation protocols, data handlers, and experiment scripts, has been submitted as supplementary material. An anonymous GitHub link is provided here: `https://anonymous.4open.science/r/TSCom-Bench-8262` and will be made public upon publication.

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

# A  APPENDIX

This appendix provides a rigorous mathematical analysis to clarify why the lossless compression evaluation paradigm offers a more comprehensive and theoretically grounded measure of a time series model's distributional modeling capabilities than the four canonical tasks of forecasting, imputation, anomaly detection, and classification.

Our central claim is that a superior generative model, parameterized by $\theta$ and denoted $Q_\theta$, should closely approximate the true data-generating distribution $P$. The gold standard for measuring the discrepancy between two probability distributions in information theory is the Kullback-Leibler (KL) divergence. An ideal evaluation metric should therefore correspond directly to minimizing $KL(P\,\|\,Q_\theta)$.

**Symbols and Definitions.** For clarity, we list all key symbols used throughout this appendix and their intended meaning (this is deliberately detailed since the appendix is read independently by reviewers):

- $X = \{x_t\}_{t=1}^T$: the original time series, each $x_t \in \mathbb{R}^d$.
- $S = f(X)$: discrete symbol sequence / byte stream produced by applying a deterministic encoding $f$ to $X$. We explicitly allow two conceptual regimes for $f$:
    1. Ideal bijection: $f$ is a one-to-one reversible mapping on the domain. In this case discrete entropies are preserved under $f$.
    2. Practical quantization: $f$ maps continuous $X$ to finite-precision representations. This mapping is many-to-one and introduces quantization error; later we quantify the information-theoretic effect.
- $P$: true distribution of $X$. In the continuous case, $p(x)$ is a probability density function (pdf) . In the discrete/bijective case, $P$ is a probability mass function (pmf).
- $Q_\theta$: model distribution over $X$ (or over symbols $S$ after applying $f$); parameterized by $\theta$.
- $x_{<t} \triangleq \{x_1, \ldots, x_{t-1}\}$: prefix / history.
- $M, O$: sets of masked and observed indices for imputation.
- $T_{\mathrm{normal}}$: indices labeled as normal for anomaly-detection training.
- $H(\cdot)$: discrete Shannon entropy in bits when argument is a pmf.
- $h(\cdot)$: differential entropy in bits when argument is a continuous density.
- $KL(P\|Q)$: Kullback–Leibler divergence, defined in the discrete case as $KL(P\|Q) = \sum_x P(x) \log_2 \frac{P(x)}{Q(x)}$, and in the continuous case as the corresponding integral when densities exist.
- All logarithms are base-2 unless otherwise noted; where natural logs appear, we indicate the conversion factor explicitly.

**Notation.** To avoid ambiguity, we distinguish three related quantities:

- Expected NLL (training loss) is the quantity minimized in training, and it equals $H(P) + KL(P\|Q_\theta)$ in the discrete case:

$$\mathcal{L}_{\mathrm{comp}}(\theta) := \mathbb{E}_{S \sim P}\big[ -\log_2 Q_\theta(S) \big]. \tag{11}$$

- Sample-level NLL is the negative log-likelihood of a particular sequence $S$ under the model:

$$\mathrm{NLL}(S) := -\log_2 Q_\theta(S), \tag{12}$$

- Arithmetic-coded length (measured file size) $L_{\mathrm{arith}}(S)$ is the actual number of bits produced by an arithmetic coder when encoding $S$ with model $Q_\theta$. By construction, $\mathrm{NLL}(S) \le L_{\mathrm{arith}}(S) < \mathrm{NLL}(S) + c$, where $c$ is a small implementation-dependent constant.

**Important conceptual distinction.** Many readers conflate: (a) theoretical statements that assume an ideal reversible encoding $f$, and (b) practical settings with finite-precision quantization. We keep these separate throughout: first state exact equalities under bijections, then provide approximations/upper bounds for practical quantization and coding.

## A.1 Invariance of Mutual Information under Bijective Mapping

A core premise of our work is that modeling the byte stream $S$ is equivalent to modeling the original continuous time series $X$. While the entropies $H(S)$ and $H(X)$ are not directly comparable, we can show that the mutual information, which captures the dependency structure, is invariant under the bijective mapping $f : X \mapsto S$.

Let's consider two continuous random vectors $X_1$ and $X_2$ with a joint probability density function (pdf) $p(x_1, x_2)$. Their mutual information is:

$$I(X_1; X_2) = \iint p(x_1, x_2) \log \frac{p(x_1, x_2)}{p(x_1)p(x_2)} dx_1 dx_2. \tag{13}$$

Now, consider a bijective (one-to-one and onto) and differentiable transformation $f$, such that $(S_1, S_2) = (f(X_1), f(X_2))$. The change of variables formula relates their pdfs:

$$q(s_1, s_2) = p(f^{-1}(s_1), f^{-1}(s_2)) \left| \det(J_{f^{-1}}(s_1, s_2)) \right|, \tag{14}$$

where $q$ is the pdf for $(S_1, S_2)$ and $J_{f^{-1}}$ is the Jacobian of the inverse transformation. The mutual information for $S_1$ and $S_2$ is:

$$I(S_1; S_2) = \iint q(s_1, s_2) \log \frac{q(s_1, s_2)}{q(s_1)q(s_2)} ds_1 ds_2. \tag{15}$$

By substituting the change of variables formula and noting that the Jacobian term cancels out in the ratio $\frac{q(s_1, s_2)}{q(s_1)q(s_2)}$, we can prove that $I(X_1; X_2) = I(S_1; S_2)$.

This invariance is critical. It implies that for our time series, the mutual information $I(x_t; x_{<t})$ is perfectly preserved. Therefore, a model that accurately learns the dependencies in the byte stream $S$ must, by extension, have learned the dependencies in the original series $X$. This provides a solid mathematical foundation for our claim that byte-level compression is a valid proxy for evaluating the modeling of continuous time series.

## A.2 On the Information Loss from Quantization

The mapping from $\mathbb{R}$ to its IEEE-754 32-bit representation is technically a form of quantization, which theoretically involves information loss. Let $X$ be the true continuous variable and $X_q$ be its quantized representation. The information loss can be quantified by the conditional differential entropy $H(X|X_q)$.

We can model quantization as adding a small, unknown error $\epsilon = X - X_q$, which is bounded by the quantization interval $\Delta$. For high-resolution quantization, it is common to approximate the error as being uniformly distributed, $\epsilon \sim U(-\Delta/2, \Delta/2)$. The entropy of this uniform distribution is $H(\epsilon) = \log_2(\Delta)$. This represents the uncertainty about the true value $X$ given its quantized version $X_q$.

In the IEEE-754 32-bit floating-point standard, the quantization step $\Delta$ is extremely small and adaptive. Most of the information lost within such tiny bins corresponds to high-frequency, unpredictable noise rather than the structured, learnable temporal patterns targeted by time series models. The signal components relevant for forecasting, imputation, or capturing seasonalities occur at a much coarser scale than the quantization resolution. Thus, while there is a theoretical information loss of approximately $\log_2(\Delta)$ bits per sample, this loss is inconsequential for the task of modeling the macroscopic statistical structure of the time series.

## A.3 Quantifying the NLL-Codelength Gap in Arithmetic Coding

Our framework relies on the fact that the achieved code length $L_{\text{arith}}(S)$ is a high-fidelity proxy for the model's sample-level negative log-likelihood, $\text{NLL}(S)$. This relationship is enabled by arithmetic coding, and we can formally analyze the gap.

There are two primary sources of sub-optimality in any practical compression scheme:

1. **Modeling Gap:** The divergence between the model's learned distribution $Q_\theta$ and the true (unknown) data distribution $P$. The expected extra code length per symbol due to this gap is the Kullback-Leibler (KL) divergence, $D_{KL}(P||Q_\theta)$. Our entire evaluation framework is designed to measure this gap.

2. **Coding Gap:** The difference between the theoretical code length prescribed by the model and the actual number of bits produced by the compressor.

An ideal entropy coder would have a coding gap of zero. Arithmetic coding is renowned for its efficiency in approaching this ideal. The extra bits redundancy of a well-implemented arithmetic coder is provably bounded. For a sequence of length $L$, the total coding gap is typically less than 2 bits for the entire sequence, arising from finite-precision arithmetic and stream termination.

$$\text{NLL}(S) \leq L_{\text{arith}}(S) < \text{NLL}(S) + c, \tag{16}$$

where $c$ is an implementation-dependent constant. The value of $c$ is typically between 1–2 bits per stream, which is an extremely tight bound. It means the contribution of the Coding Gap to the final file size is negligible. Therefore, the measured compressed length $L_{\text{comp}}$ is almost entirely determined by the model's NLL. This validates our use of the final compressed size as a direct and stringent measure of the model's probabilistic modeling capability.

### A.4 LOSSLESS COMPRESSION: THE GOLD STANDARD

We keep your original derivation and expand each step with a full explanation.

For a time series $X = \{x_t\}_{t=1}^T$, assume an autoregressive factorization of the model distribution:

$$Q_\theta(X) = \prod_{t=1}^T Q_\theta(x_t \mid x_{<t}). \tag{17}$$

The compression loss is the expected NLL:

$$\mathcal{L}_{\text{comp}}(\theta) = \mathbb{E}_{X \sim P}[-\log_2 Q_\theta(X)]. \tag{18}$$

Now reproduce and expand your original algebraic decomposition:

$$
\begin{aligned}
\mathcal{L}_{\text{comp}}(\theta) &= \mathbb{E}_{X \sim P}[-\log_2 Q_\theta(X)] \\
&= \mathbb{E}_{X \sim P}\Big[ -\log_2 P(X) + \log_2 \frac{P(X)}{Q_\theta(X)}\Big] \\
&= \mathbb{E}_{X \sim P}[-\log_2 P(X)] + \mathbb{E}_{X \sim P}\Big[\log_2 \frac{P(X)}{Q_\theta(X)}\Big] \\
&= H(P) + KL(P\,||\,Q_\theta). 
\end{aligned} \tag{19}
$$

As shown in equation 19, the first equality is the definition of expected NLL under $P$. In the second line, we add and subtract $\log_2 P(X)$ inside the expectation. This is an exact algebraic identity:

$$-\log_2 Q_\theta(X) = -\log_2 P(X) + \log_2 \frac{P(X)}{Q_\theta(X)}. \tag{20}$$

Then the third line separates the expectation over the sum into the sum of expectations. The fourth line recognizes $\mathbb{E}_{X \sim P}[-\log_2 P(X)]$ as the Shannon entropy $H(P)$ (in bits), and $\mathbb{E}_{X \sim P}\big[\log_2 \frac{P(X)}{Q_\theta(X)}\big]$ as the Kullback–Leibler divergence $KL(P||Q_\theta)$. Therefore, the information is important for clarification:

1. Since $H(P)$ depends only on the true distribution $P$, it is a constant with respect to model parameters $\theta$. Therefore minimizing $\mathcal{L}_{\text{comp}}(\theta)$ is equivalent to minimizing $KL(P||Q_\theta)$.

2. The above equality is exact for discrete distributions where pmfs exist. For continuous-valued $X$ with densities, the analogous decomposition holds if $P$ and $Q_\theta$ admit densities w.r.t. the same dominating measure. Otherwise, one must work in terms of measures.

3. The metric $KL(P||Q_\theta)$ is global: it penalizes all deviations of $Q_\theta$ from $P$, including differences in support, modes, tails, and higher moments, which explains why compression is a strict measure of distributional fit.

**Gradient form.** It is often useful to see the gradient of the compression loss:

$$
\begin{aligned}
\nabla_\theta \mathcal{L}_{\text{comp}}(\theta) &= \nabla_\theta \mathbb{E}_{X \sim P}[-\log_2 Q_\theta(X)] \\
&= -\mathbb{E}_{X \sim P}\big[\nabla_\theta \log_2 Q_\theta(X)\big] \\
&= -\frac{1}{\ln 2} \mathbb{E}_{X \sim P}\big[\nabla_\theta \ln Q_\theta(X)\big],
\end{aligned}
\tag{21}
$$

where we used $\log_2 u = (\ln u)/\ln 2$. This shows that training under $\mathcal{L}_{\text{comp}}(\theta)$ provides gradient signals from every $X$ sampled from $P$, in contrast to restricted losses.

**Practical coding: arithmetic coding and finite-precision overhead.** When using arithmetic coding to convert model probabilities into bitstreams, the achieved code length for a sequence $S$ satisfies:

$$
\text{NLL}(S) \ \leq \ L_{\text{arith}}(S) \ < \ \text{NLL}(S) + c,
\tag{22}
$$

where $c$ is a small implementation-dependent constant (Cover & Thomas, 2006). Hence asymptotically, the NLL is an achievable lower bound on practical codelength up to a negligible constant overhead.

## A.5 Comparison with Canonical Tasks

We provide a detailed comparison between lossless compression and the four canonical evaluation tasks widely used in time series modeling: forecasting, imputation, anomaly detection, and classification.

**Forecasting.** Forecasting aims to predict the future values given the past. The standard loss is mean squared error:

$$
\mathcal{L}_{\text{forecast}}(\theta) = \frac{1}{T} \sum_{t=1}^{T} \|x_t - \hat{x}_t^\theta\|_2^2, \quad \hat{x}_t^\theta = \mathbb{E}_{Q_\theta}[x_t \mid x_{<t}].
\tag{23}
$$

Minimizing this loss forces $Q_\theta$ to match only the conditional mean. Different distributions can share the same mean but have very different variance or tail behaviour, so a model may achieve low forecasting loss yet diverge from $P$ in KL divergence.

**Imputation.** Imputation requires the model to reconstruct missing values in a partially observed sequence. Let $M \subset \{1, \ldots, T\}$ be a randomly sampled set of masked indices, and let $O$ denote the complement set of observed indices. A typical objective is to minimize the mean squared error on the masked values, denoted by $\mathcal{L}_{\text{imp}}$:

$$
\mathcal{L}_{\text{imp}}(\theta) = \mathbb{E}_M \left[ \sum_{t \in M} \big\|x_t - \hat{x}_t^\theta(x_O)\big\|_2^2 \right],
\tag{24}
$$

where the expectation $\mathbb{E}_M$ is taken over the distribution of masks, and $\hat{x}_t^\theta(x_O)$ is the model's reconstruction of $x_t$ conditioned on the observed values $x_O$. This criterion enforces local accuracy only on masked positions, while unmasked positions are unconstrained. Unless masking covers all possible subsets, $Q_\theta$ can match $\mathcal{L}_{\text{imp}}$ while disagreeing with $P$ elsewhere.

**Anomaly detection.** The model learns the density of normal data and flags deviations. A common approach is to maximize the likelihood on the set of normal data points. Let $T_{\text{normal}} \subset \{1, \ldots, T\}$ be

the set of time indices corresponding to normal data. The loss $\mathcal{L}_{\text{anom}}$ is the negative log-likelihood on this subset:

$$\mathcal{L}_{\text{anom}}(\theta) = - \sum_{t \in T_{\text{normal}}} \log_2 Q_\theta(x_t \mid x_{<t}). \qquad (25)$$

This objective enforces accurate density estimation only within the restricted support of normal sequences. Probability mass outside this region is largely irrelevant, meaning the model is not penalized for misrepresenting the full distribution.

**Classification.** Classification associates an entire sequence $X$ with a single, discrete label $y \in \mathcal{Y}$, where $\mathcal{Y}$ is the set of all possible labels. The standard objective is to minimize the cross-entropy loss, denoted by $\mathcal{L}_{\text{cls}}$:

$$\mathcal{L}_{\text{cls}}(\theta) = - \log_2 Q_\theta(y \mid X). \qquad (26)$$

This objective enforces that the model's conditional label distribution $Q_\theta(y \mid X)$ approximates the true one $P(y \mid X)$, but it does not constrain the sequence distribution $Q_\theta(X)$ itself. A model may achieve perfect classification by exploiting only a few discriminative features, while ignoring most temporal dependencies.

### A.6 Forecasting: Constraining Only the Conditional Mean

Forecasting tasks typically employ the Mean Squared Error (MSE) loss:

$$\mathcal{L}_{\text{forecast}}(\theta) = \mathbb{E}_{X \sim P} \left[ \frac{1}{T} \sum_{t=1}^{T} \| x_t - \hat{x}_t^\theta \|_2^2 \right] \qquad (27)$$

where the point forecast $\hat{x}_t^\theta$ is the conditional expectation under the model: $\hat{x}_t^\theta := \mathbb{E}_{Q_\theta}[x_t | x_{<t}]$.

**Mathematical Derivation and Analysis.** To minimize $\mathcal{L}_{\text{forecast}}$, for any given history $x_{<t}$, the model must select an optimal prediction $\hat{x}_t$ that minimizes the expected squared error under the true conditional distribution $P(x_t | x_{<t})$. We find this optimal point by taking the derivative with respect to $\hat{x}_t$ and setting it to zero:

$$\frac{\partial}{\partial \hat{x}_t} \mathbb{E}_{P(x_t | x_{<t})} [\| x_t - \hat{x}_t \|_2^2] = \mathbb{E}_{P(x_t | x_{<t})} \left[ \frac{\partial}{\partial \hat{x}_t} (x_t - \hat{x}_t)^T (x_t - \hat{x}_t) \right]$$
$$= \mathbb{E}_{P(x_t | x_{<t})} [-2(x_t - \hat{x}_t)]$$
$$= -2(\mathbb{E}_{P(x_t | x_{<t})}[x_t] - \hat{x}_t) \qquad (28)$$

Setting the derivative to zero yields the optimal forecast $\hat{x}_t^{\text{opt}} = \mathbb{E}_{P(x_t | x_{<t})}[x_t]$. This derivation proves that minimizing the MSE loss solely drives the mean of the model's predictive distribution, $\mathbb{E}_{Q_\theta}[x_t | x_{<t}]$, to match the mean of the true conditional distribution.

**Comparison with Compression.** The MSE objective is limited as it only constrains the first moment of the distribution, while remaining insensitive to all higher-order moments and the overall distributional shape. A model can achieve a perfect MSE score with a unimodal Gaussian prediction, even if the true distribution is bimodal, leading to a potentially infinite KL divergence.

### A.7 Imputation: Constraining a Subset of Conditional Means

The imputation loss is also typically an MSE objective:

$$\mathcal{L}_{\text{imp}}(\theta) = \mathbb{E}_M \left[ \sum_{t \in M} \| x_t - \hat{x}_t^\theta(x_O) \|_2^2 \right] \qquad (29)$$

where $M$ is the set of masked indices, $O$ is the set of observed indices, and $\hat{x}_t^\theta(x_O) := \mathbb{E}_{Q_\theta}[x_t | x_O]$.

**Mathematical Derivation and Analysis.** To minimize this loss, for any given set of observed values $x_O$, the model must find the optimal imputation $\hat{x}_t(x_O)$ that minimizes the expected squared

error under the true conditional distribution $P(x_t|x_O)$. We derive this optimal value by taking the derivative with respect to $\hat{x}_t(x_O)$ and setting it to zero:

$$\frac{\partial}{\partial \hat{x}_t(x_O)} \mathbb{E}_{P(x_t|x_O)}[\|x_t - \hat{x}_t(x_O)\|_2^2]$$

$$= \mathbb{E}_{P(x_t|x_O)}\left[\frac{\partial}{\partial \hat{x}_t(x_O)}(x_t - \hat{x}_t(x_O))^T(x_t - \hat{x}_t(x_O))\right]$$

$$= \mathbb{E}_{P(x_t|x_O)}[-2(x_t - \hat{x}_t(x_O))]$$

$$= -2(\mathbb{E}_{P(x_t|x_O)}[x_t] - \hat{x}_t(x_O)) \tag{30}$$

Setting the final expression to zero yields the optimal imputation:

$$\hat{x}_t^{\text{opt}}(x_O) = \mathbb{E}_{P(x_t|x_O)}[x_t] \tag{31}$$

This derivation formally shows that minimizing the imputation loss solely forces the model's conditional mean, $\mathbb{E}_{Q_\theta}[x_t|x_O]$, to align with the true conditional mean.

**Comparison with Compression.** This derivation highlights two fundamental limitations: (1) Like forecasting, it only constrains the conditional mean, ignoring the full conditional distribution $P(x_M|x_O)$. (2) The objective is optimized only over a specific masking strategy, offering no guarantee that the model has learned the full joint distribution $P(X)$ required to handle arbitrary patterns of missingness. Compression, by contrast, requires modeling all conditionals $P(x_t|x_{<t})$ and thus captures the full joint distribution.

### A.8 ANOMALY DETECTION: CONSTRAINING LIKELIHOOD ON A RESTRICTED SUPPORT

A common anomaly-detection training objective is to maximize (or equivalently minimize negative) likelihood over normal data only:

$$\mathcal{L}_{\text{anom}}(\theta) = -\sum_{t \in T_{\text{normal}}} \log_2 Q_\theta(x_t \mid x_{<t}). \tag{32}$$

**Gradient-level analysis.** The gradient of this objective is

$$\nabla_\theta \mathcal{L}_{\text{anom}}(\theta) = -\sum_{t \in T_{\text{normal}}} \nabla_\theta \log_2 Q_\theta(x_t \mid x_{<t})$$

$$= -\frac{1}{\ln 2} \sum_{t \in T_{\text{normal}}} \frac{\nabla_\theta Q_\theta(x_t \mid x_{<t})}{Q_\theta(x_t \mid x_{<t})}. \tag{33}$$

Only indices in $T_{\text{normal}}$ contribute to the gradient; anomalous samples do not appear and thus provide no direct learning signal.

**Implication.** Because anomalies are absent from the training gradient, the model is not explicitly encouraged to give them low probability, which is only encouraged to give high probability to normal examples. A model could, in principle, assign arbitrarily large probability mass to certain anomalous patterns while still maximizing the objective on normal data. In contrast, the compression objective enforces low likelihood for rare/unexpected events insofar as assigning mass to those events increases expected code length.

**Comparison with Compression.** The gradient analysis proves that the model receives no supervision on how to assign probabilities to anomalous events. The model is not penalized for assigning high probability to anomalies, which fundamentally undermines its ability to detect them. The compression objective $\mathcal{L}_{\text{comp}}(\theta)$ computes the NLL over all data points ($t = 1, \ldots, T$), ensuring that its gradient reflects the need to assign low probability to rare events to achieve an efficient overall codelength.

### A.9 CLASSIFICATION: CONSTRAINING ONLY THE LABEL'S POSTERIOR PROBABILITY

The classification objective is to minimize the cross-entropy loss:

$$\mathcal{L}_{\text{cls}}(\theta) = -\log_2 Q_\theta(y|X) \tag{34}$$

**Mathematical Derivation and Analysis.** The expected loss over the true data distribution $P(X, Y)$ is:

$$\mathbb{E}_{(X,y)\sim P(X,Y)}[-\log_2 Q_\theta(y|X)] = \sum_{X,y} P(X, y)[-\log_2 Q_\theta(y|X)]$$

$$= \sum_{X,y} P(X, y)\left[-\log_2 P(y|X) + \log_2 \frac{P(y|X)}{Q_\theta(y|X)}\right]$$

$$= H(Y|X) + KL(P(Y|X) \,\|\, Q_\theta(Y|X)) \tag{35}$$

where $H(Y|X)$ is the true conditional entropy of the labels given the data, a constant with respect to the model. This derivation formally shows that the classification objective is solely concerned with minimizing the KL divergence between the true conditional label distribution $P(Y|X)$ and the model's prediction $Q_\theta(Y|X)$.

**Comparison with Compression.** The joint distribution of data and labels is $P(X, Y) = P(Y|X)P(X)$. The mathematics clearly shows that the classification objective focuses exclusively on the $P(Y|X)$ term and places absolutely no constraints on the modeling of the data distribution $P(X)$ itself. A model can achieve perfect classification by learning a mapping from a small, discriminative subset of features in $X$ to $y$, while completely failing to capture the underlying generative process of $X$. Compression, in contrast, directly evaluates the model's understanding of $P(X)$, making the two objectives mathematically orthogonal.

## A.10 Unified View and Summary

The analyses above show that the four canonical tasks evaluate a model by minimizing a divergence on a "projection" or "subset" of the true data distribution. We summarize this in Table 6.

Table 6: Unified Mathematical View of Evaluation Tasks

| Task | Objective Function $\mathcal{L}_{\text{task}}$ | Optimized Statistic/Distribution $\phi(\cdot)$ | Key Mathematical Limitation |
|---|---|---|---|
| Compression | $\mathcal{L}_{\text{comp}}(\theta)$ | **Full Distribution** $P(X)$ | None (Theoretically global evaluation) |
| Forecasting | $\mathbb{E}_P[\|x_t - \hat{x}_t^\theta\|_2^2]$ | **Conditional Mean** $\mathbb{E}[x_t|x_{<t}]$ | Ignores all higher-order moments and shape |
| Imputation | $\mathbb{E}_M[\|x_t - \hat{x}_t^\theta(x_O)\|_2^2]$ | **Subset of Cond. Means** $\mathbb{E}[x_t|x_O]$ | Constrains only the mean; depends on mask strategy |
| Anomaly Det. | $-\sum_{t\in T_{\text{normal}}} \log_2 Q_\theta(x_t|x_{<t})$ | **Dist. on a Subset** $P(X)|_{X\in\text{Normal}}$ | No constraint on probability of anomalous events |
| Classification | $-\log_2 Q_\theta(y|X)$ | **Label Posterior Dist.** $P(y|X)$ | No constraint on the data distribution $P(X)$ |

In conclusion, the mathematical derivations confirm that lossless compression, by being equivalent to minimizing the full KL divergence, provides a holistic, unified, and strict evaluation of a model's generative capabilities. The canonical tasks, in contrast, examine only specific, and often insufficient, aspects of the true data distribution.

## A.11 Overview of Core Process of Arithmetic Encoding

The arithmetic encoder processes byte stream data (with a symbol set of discrete symbols ranging from 0 to 255) based on its core principle of interval mapping for data compression: it maps the original byte sequence to a continuous decimal number within the interval [0,1], which is then represented by the shortest binary form to generate the compressed bitstream. During decoding, the probability distribution from the encoding end is reused to iteratively restore the original symbol sequence through reverse operations. The encoder's performance relies on two key logical components: first, cumulative probability modeling, which converts the probability distribution of bytes into exclusive subintervals within [0,1), assigning each byte a unique interval range; second, iterative interval reduction, where the current interval is subdivided using the exclusive subinterval of the current symbol during encoding, and symbols are located via interval matching during decoding. Both processes share identical interval update rules to ensure lossless data reconstruction. Next, we will elaborate on the core workflow of arithmetic encoding in three stages.

**Construction of Cumulative Probability Distribution** The core input of arithmetic encoding is not individual probabilities, but the cumulative probability distribution. Because it requires partitioning the interval $[0, 1)$ using cumulative probabilities to assign each byte a unique subinterval. This conversion serves as the bridge connecting the model's output and the encoding operation.

First, clarify the form of the time-series model's output: assume the model predicts the probability distribution of the next byte as $P = [p_0, p_1, \ldots, p_{255}]$, where $p_i$ is the probability that the next byte equals $i$ ($0 \leq i \leq 255$), satisfying $\sum_{i=0}^{255} p_i = 1$. Then, define a cumulative probability array $C = [C_0, C_1, \ldots, C_{256}]$ of length 257, covering the start and end points of intervals for bytes 0 to 255. Initialize $C_0 = 0$ (the starting baseline), and compute subsequent elements through cumulative probability summation:

$$C_{i+1} = C_i + p_i \tag{36}$$

Ultimately, $C_{256} = 1$, ensuring full coverage of the interval. Through this process, the exclusive interval for byte $i$ is $[C_i, C_{i+1})$, with an interval width equal to its probability $p_i$, aligning with the compression principle of assigning wider intervals to high-frequency bytes and narrower intervals to low-frequency bytes. For example, suppose the model outputs a set of values as shown in the Table 7. Bytes with higher probabilities are assigned longer intervals, which is the key to subsequent short encoding.

Table 7: Byte Probability Distribution and Interval Partitioning

| Byte $i$ | Probability $p_i$ | Cumulative Probability $c_i$ | Cumulative Probability $c_{i+1}$ | Exclusive Interval for Byte $i$ | Interval Length ($= p_i$) |
|---|---|---|---|---|---|
| 0-107 | Sum 0.1 | 0.0 | 0.1 | $[0.0, 0.1)$ | 0.1 |
| 108 | 0.15 | 0.1 | 0.25 | $[0.1, 0.25)$ | 0.15 |
| 109-113 | Sum 0.1 | 0.25 | 0.35 | $[0.25, 0.35)$ | 0.1 |
| 114 | 0.45 (Target Byte) | 0.35 | 0.8 | $[0.35, 0.8)$ | 0.45 |
| 115-255 | Sum 0.2 | 0.8 | 1.0 | $[0.8, 1.0)$ | 0.2 |

**Narrow down the encoding range using the actual byte's interval**   The essence of arithmetic encoding lies in progressively narrowing the interval and using the final interval's binary representation as the encoding result. The narrowing process is guided by the model's assigned exclusive interval for each byte. Specifically, for encoding the actual byte 114, let the initial encoding interval be $[0, 1)$. When the actual byte is 114, we use its exclusive interval $[0.35, 0.8)$ to carve the current encoding interval $[0, 1)$, resulting in a new encoding interval $[0.35, 0.8)$.

Table 8: The Structure of Binary Sub-intervals for Final Code Selection. This table illustrates how binary fractions of varying lengths (precision) partition the unit interval $[0, 1)$. This principle is used in the final step of arithmetic encoding to select the shortest binary code that uniquely represents a sub-interval contained entirely within the algorithm's final target range.

| Binary Decimal Digits | Division Precision (Interval Length) | Interval Examples (Partial) | Meaning of Binary Fractions |
|---|---|---|---|
| 1-digit ($0.x_1$) | $1/2 = 0.5$ | $[0, 0.5), [0.5, 1)$ | $0.1 \to [0.5, 1), 0.0 \to [0, 0.5)$ |
| 2-digit ($0.x_1 x_2$) | $1/4 = 0.25$ | $[0, 0.25), [0.25, 0.5), \ldots$ | $0.10 \to [0.5, 0.75)$ |
| 3-digit ($0.x_1 x_2 x_3$) | $1/8 = 0.125$ | $[0, 0.125), [0.125, 0.25), \ldots$ | $0.101 \to [0.625, 0.75)$ |
| $n$-digit | $1/2^n$ | $[k/2^n, (k+1)/2^n)$ ($k = 0, 1, \ldots, 2^n - 1$) | $n$-digit binary fractions correspond to intervals of length $1/2^n$ |

**Final Encoded Output**   The ultimate goal of the encoding process is to use a sequence of binary bits to uniquely represent this interval. For instance, the shortest binary fraction serves as an efficient representation, and any two distinct binary fractions must correspond to different numerical values, thereby satisfying the prerequisite of encoding uniqueness. For the new interval $(0.35, 0.8)$, we seek the shortest binary fraction such that its corresponding subinterval entirely falls within $(0.35, 0.8)$. As shown in the Table 8, among 2-bit binary fractions, $0.10_2$ (corresponding to the decimal value 0.5) has a subinterval of $(0.5, 0.75)$, which lies entirely within $(0.35, 0.8)$. Thus, the encoding result is 1 0, using only 2 bits, which is significantly fewer than the traditional 8-bit encoding.

### A.12 ADDITIONAL EXPERIMENTS

To further validate the effectiveness and robustness of our proposed lossless compression evaluation paradigm, we conduct additional experiments on a diverse set of benchmark datasets. In this appendix, we provide detailed descriptions of each dataset, the parameter settings used in our experiments, and the full results under multiple sequence lengths. This section complements the main text by reporting comprehensive results that could not fit within the page limits.

#### A.12.1 DATASETS

We evaluate on six widely-used public datasets covering diverse application domains:

- **PEMS04 and PEMS08** are traffic flow datasets collected from the California Department of Transportation's Performance Measurement System. They contain traffic speed, flow, and occupancy data from hundreds of loop sensors on highway networks. We follow standard preprocessing and use the same train, validation and test splits as prior works.
- **Traffic** contains road occupancy rates measured by 862 sensors on San Francisco Bay Area freeways. It is a canonical benchmark for large-scale multivariate time series forecasting.
- **Electricity** records hourly electricity consumption of 321 customers from 2012–2014. It exhibits strong daily and weekly periodicity, making it a challenging testbed for temporal models.
- **Weather** contains 21 meteorological variables collected from the WeatherBench benchmark. It is commonly used to evaluate long-horizon temporal modeling under rich covariates.
- **ETTh2 and ETTm2** are subsets of the ETT (Electricity Transformer Temperature) benchmark capturing transformer oil temperature and related exogenous factors. ETTh2 has hourly resolution, while ETTm2 has 15-minute resolution, enabling evaluation across different temporal granularities.

We also include several standard lossless compression benchmarks to evaluate the general-purpose capabilities of the models:

- **Enwik9** is a standard benchmark from the Large Text Compression Benchmark, consisting of the first 1 billion bytes of an English Wikipedia XML dump. It is widely used to test a compressor's performance on natural language text.
- **Image** is a dataset composed of raw, uncompressed image bitmaps derived from the ImageNet database, designed to evaluate compression performance on visual data with high spatial redundancy.
- **Sound** consists of uncompressed audio waveforms from environmental sound recordings, which tests a model's ability to capture the temporal structures and periodic patterns typical in audio data.
- **Float** is a dataset containing arrays of 64-bit double-precision floating-point numbers from scientific simulations. It is used to benchmark the compression of high-precision numerical data.
- **Silesia Corpus** is a well-known collection of diverse file types, including text, executables, images, and databases, designed to be a representative benchmark for general-purpose lossless compressors.
- **Backup** is a heterogeneous dataset created to simulate real-world backup archives, containing a mixture of different file types to test a compressor's ability to adapt to varying data statistics.

#### A.12.2 PARAMETERS AND EXPERIMENTAL SETUP

The experiments in this section follow the setup described in the main paper. We evaluate a suite of eight representative time series models on six public benchmarks. To assess performance robustness, we test across four distinct sequence lengths: {12, 24, 48, 96}. All reported results are averaged over three independent runs with different random seeds to ensure reliability.

Table 9: Comprehensive lossless compression results across six benchmark datasets under multiple sequence lengths. The best result in each setting is highlighted in **bold**, the second best is underlined, and *avg* denotes the average over all tested horizons.

| Dataset | Horizon | TimeXer (2025) CR | TimeXer (2025) CT | iTransformer (2024) CR | iTransformer (2024) CT | PatchTST (2023) CR | PatchTST (2023) CT | Autoformer (2023) CR | Autoformer (2023) CT | DLinear (2023) CR | DLinear (2023) CT | LightTS (2023) CR | LightTS (2023) CT | SCINet (2022) CR | SCINet (2022) CT | Informer (2021) CR | Informer (2021) CT |
|---|---|---|---|---|---|---|---|---|---|---|---|---|---|---|---|---|---|
| PEMS08 | 12 | 0.979 | 12.42 | **0.977** | 23.34 | 0.980 | 25.12 | 0.980 | 3.33 | 0.998 | **33.13** | 0.985 | 24.45 | 0.994 | 2.36 | 0.993 | 2.85 |
| | 24 | 0.979 | 16.23 | **0.976** | 24.19 | 0.979 | 21.88 | 0.980 | 4.42 | 0.998 | 32.54 | 0.986 | 24.38 | 0.983 | 2.17 | 0.982 | 2.73 |
| | 48 | 0.979 | 16.00 | **0.978** | 24.06 | 0.978 | 16.23 | 0.979 | 1.28 | 0.997 | 30.33 | 0.991 | 23.62 | 0.989 | 1.98 | 0.980 | 2.66 |
| | 96 | **0.978** | 12.55 | 0.978 | 18.13 | 0.978 | 9.63 | 0.980 | 3.24 | 0.996 | 30.92 | 0.989 | 17.41 | 0.980 | 2.74 | 0.979 | 2.74 |
| | avg | 0.979 | 14.30 | **0.978** | 22.43 | 0.979 | 18.22 | 0.980 | 3.07 | 0.997 | **31.73** | 0.988 | 22.47 | 0.987 | 2.31 | 0.984 | 2.75 |
| Traffic | 12 | **0.139** | 21.36 | 0.141 | 35.89 | 0.139 | 38.22 | 0.171 | 3.59 | 0.158 | **60.74** | 0.146 | 35.33 | 0.357 | 1.33 | 0.191 | 4.55 |
| | 24 | **0.138** | 20.44 | 0.140 | 35.28 | 0.138 | 29.46 | 0.164 | 4.86 | 0.154 | 62.66 | 0.180 | 35.15 | 0.159 | 1.78 | 0.172 | 3.94 |
| | 48 | **0.137** | 20.23 | 0.141 | 36.35 | 0.137 | 20.62 | 0.162 | 1.30 | 0.153 | 59.87 | 0.174 | 33.31 | 0.158 | 1.27 | 0.166 | 4.24 |
| | 96 | **0.137** | 15.58 | 0.141 | 23.21 | 0.137 | 11.89 | 0.151 | 3.27 | 0.155 | 60.06 | 0.174 | 24.63 | 0.140 | 1.29 | 0.167 | 4.18 |
| | avg | **0.138** | 19.40 | 0.141 | 32.68 | 0.138 | 25.05 | 0.162 | 3.26 | 0.155 | **60.83** | 0.169 | 32.11 | 0.204 | 1.42 | 0.174 | 4.23 |
| Electricity | 12 | **0.131** | 20.96 | 0.132 | 35.49 | 0.131 | 38.10 | 0.157 | 3.59 | 0.178 | **64.11** | 0.168 | 34.67 | 0.216 | 2.46 | 0.209 | 3.97 |
| | 24 | **0.128** | 20.78 | 0.133 | 36.06 | 0.128 | 29.74 | 0.185 | 4.87 | 0.173 | **65.14** | 0.180 | 35.36 | 0.205 | 2.72 | 0.211 | 4.01 |
| | 48 | **0.119** | 20.45 | 0.134 | 35.75 | 0.121 | 21.28 | 0.267 | 1.31 | 0.173 | 63.41 | 0.172 | 33.09 | 0.168 | 2.77 | 0.202 | 4.13 |
| | 96 | **0.112** | 16.19 | 0.142 | 23.06 | 0.115 | 12.32 | 0.194 | 3.26 | 0.176 | 57.79 | 0.168 | 24.33 | 0.135 | 2.87 | 0.194 | 4.17 |
| | avg | **0.123** | 19.60 | 0.135 | 32.59 | 0.124 | 25.36 | 0.201 | 3.26 | 0.175 | **62.61** | 0.172 | 31.86 | 0.181 | 2.71 | 0.204 | 4.07 |
| Weather | 12 | **0.229** | 20.13 | 0.236 | 34.30 | 0.234 | 35.78 | 0.344 | 3.52 | 0.418 | **53.53** | 0.379 | 29.69 | 0.497 | 3.12 | 0.482 | 3.11 |
| | 24 | **0.209** | 20.02 | 0.217 | 33.49 | 0.212 | 28.77 | 0.356 | 4.75 | 0.388 | 53.12 | 0.377 | 29.94 | 0.367 | 2.99 | 0.451 | 2.78 |
| | 48 | **0.208** | 20.01 | 0.296 | 29.08 | 0.211 | 20.67 | 0.359 | 1.30 | 0.382 | 56.08 | 0.384 | 31.54 | 0.343 | 3.53 | 0.424 | 2.83 |
| | 96 | **0.207** | 15.63 | 0.268 | 21.99 | 0.213 | 11.76 | 0.370 | 2.15 | 0.382 | 54.57 | 0.370 | 20.56 | 0.332 | 3.52 | 0.418 | 2.77 |
| | avg | **0.213** | 18.95 | 0.254 | 29.72 | 0.218 | 24.25 | 0.357 | 2.93 | 0.393 | 54.33 | 0.378 | 27.93 | 0.385 | 3.29 | 0.444 | 2.87 |
| ETTh2 | 12 | **0.267** | 19.79 | 0.277 | 33.74 | 0.279 | 32.80 | 0.393 | 3.51 | 0.541 | **48.62** | 0.520 | 29.16 | 0.499 | 3.02 | 0.493 | 2.73 |
| | 24 | **0.260** | 19.00 | 0.279 | 30.67 | 0.274 | 27.14 | 0.423 | 4.73 | 0.504 | 47.36 | 0.488 | 29.52 | 0.484 | 3.12 | 0.478 | 2.61 |
| | 48 | **0.267** | 19.15 | 0.303 | 31.07 | 0.279 | 20.01 | 0.426 | 1.32 | 0.491 | 51.45 | 0.536 | 28.84 | 0.443 | 2.97 | 0.438 | 2.82 |
| | 96 | **0.262** | 15.04 | 0.364 | 20.50 | 0.285 | 11.67 | 0.404 | 2.17 | 0.495 | 44.72 | 0.534 | 22.13 | 0.412 | 3.53 | 0.437 | 2.74 |
| | avg | **0.264** | 18.25 | 0.306 | 29.00 | 0.279 | 22.91 | 0.412 | 2.93 | 0.508 | 48.04 | 0.520 | 27.41 | 0.460 | 3.16 | 0.462 | 2.73 |
| Solar | 12 | 0.073 | 21.57 | **0.064** | 37.28 | 0.075 | 38.37 | 0.093 | 3.55 | 0.104 | **64.92** | 0.081 | 38.78 | 0.078 | 2.31 | 0.101 | 2.33 |
| | 24 | **0.025** | 21.28 | 0.030 | 38.43 | 0.028 | 31.38 | 0.088 | 4.53 | 0.074 | 69.32 | 0.054 | 40.47 | 0.064 | 2.73 | 0.098 | 2.63 |
| | 48 | **0.025** | 21.42 | 0.035 | 36.52 | 0.028 | 21.98 | 0.078 | 1.33 | 0.067 | 70.63 | 0.053 | 36.68 | 0.053 | 2.87 | 0.094 | 2.86 |
| | 96 | **0.027** | 16.61 | 0.036 | 24.70 | 0.029 | 21.98 | 0.074 | 2.79 | 0.068 | 65.55 | 0.055 | 27.22 | 0.049 | 2.90 | 0.093 | 2.79 |
| | avg | **0.038** | 20.22 | 0.041 | 34.23 | 0.040 | 28.43 | 0.083 | 3.05 | 0.078 | **67.61** | 0.061 | 35.79 | 0.061 | 2.70 | 0.097 | 2.65 |

### A.12.3 FULL LOSSLESS COMPRESSION RESULTS AND ANALYSIS

An analysis of Table 9 reveals several key findings. Overall, TimeXer consistently achieves the best or second-best CR across nearly all datasets and horizons, affirming its strong capability in capturing data distributions. iTransformer and PatchTST also demonstrate highly competitive compression performance, often securing top-tier rankings. Beyond absolute performance, the results highlight a clear rate-utility trade-off. For example, the simple linear model DLinear exhibits by far the highest CT, making it the fastest method, but this speed comes at the cost of a significantly poorer compression ratio. Conversely, models like TimeXer provide superior compression with more moderate throughput, showcasing how the benchmark can quantify this critical trade-off. The benchmark's validity is further validated by its ability to characterize datasets: the PEMS08 dataset consistently yields a CR close to 1.0, correctly identifying its pre-compressed nature, while the highly predictable Solar dataset results in very low CR values. Collectively, these detailed results reinforce the importance of lossless compression as a robust and insightful evaluation paradigm. It moves beyond single-purpose metrics to provide a multi-faceted view of a model's performance, assessing not only its fundamental ability to model data distributions but also its practical trade-offs regarding speed and sensitivity to data characteristics.

Surprisingly, the Solar dataset exhibits extraordinarily strong compressibility: under the TimeXer model, the compressed file size is approximately 3% of the original. To understand this behaviour, we performed a dataset-level diagnostic (Fig. 4). The panel (a) shows that 55.10% of all entries are

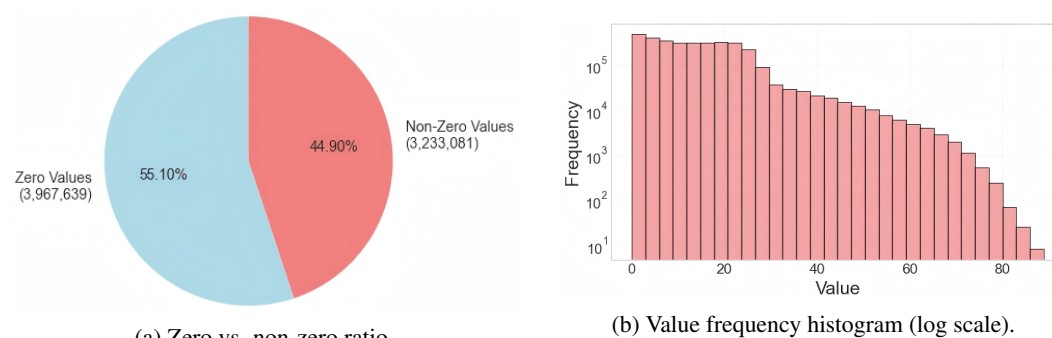

(a) Zero vs. non-zero ratio.

(b) Value frequency histogram (log scale).

Figure 4: Dataset diagnostics explaining Solar's exceptional compressibility and PEMS08's apparent incompressibility. (a) shows that 55.10% of Solar entries are exactly zero; (b) shows a strongly skewed value-frequency distribution with only 2,539 unique values over 7,200,720 samples and values confined to $[0.0, 88.9]$. These properties make Solar highly predictable for neural compressors.

exactly zero, producing long runs of highly predictable values. The panel (b) reveals that the dataset contains 7,200,720 samples but only 2,539 unique values (a repetition rate of roughly 99.96%), with non-zero values confined to a narrow numeric range $[0.0, 88.9]$. These characteristics—high sparsity, extreme redundancy, a limited numeric range, and pronounced diurnal/seasonal periodicity—concentrate probability mass and make the series especially easy for neural autoregressive predictors to model accurately, which in turn yields very low bits-per-byte and excellent compression. By contrast, the apparently poor compressibility of PEMS08 is an artifact of its storage format: PEMS08 is distributed as a `.npz` archive, so the files are already compressed and contain little residual redundancy for further reduction, producing compression ratios close to one.

### A.12.4 ANALYSIS OF COMPRESSION DYNAMICS AND MODEL CONVERGENCE

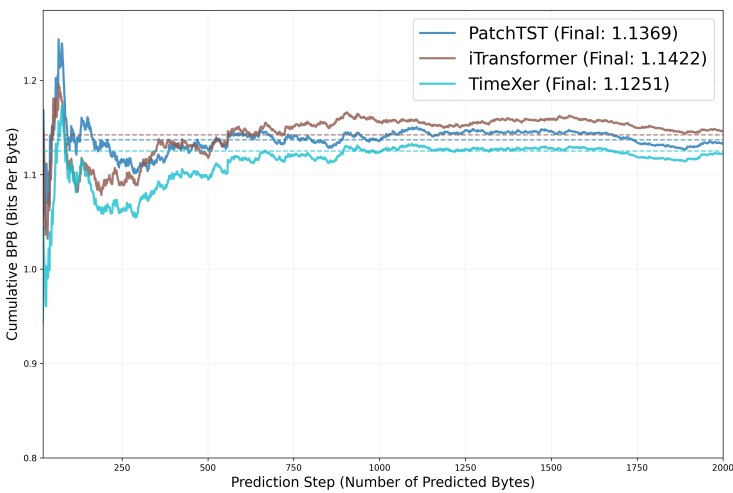

Figure 5: Step-by-step convergence of cumulative bpb for top-performing models on the synthetic dataset. The legend reports the final, stable BPB value achieved by each model after processing 2,000 bytes.

To provide deeper insight into the compression process, we visualize the step-by-step performance of our top models on the synthetic dataset. Figure 5 plots the cumulative bpb as a function of the number of bytes processed. The cumulative bpb acts as a running average of compression efficiency, reflecting how well the model predicts the data stream over time.

The plot reveals several key behaviors. Initially, the bpb for all models is volatile, which is expected when the predictive context is small. However, as the models process more data, their performance

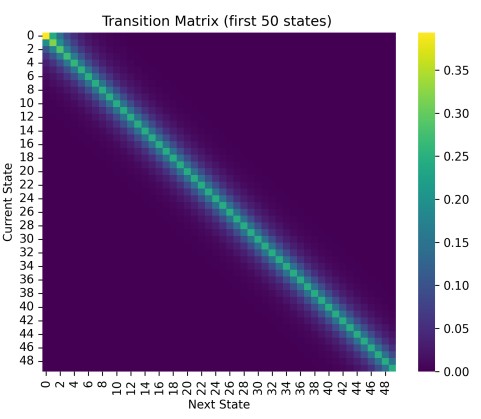
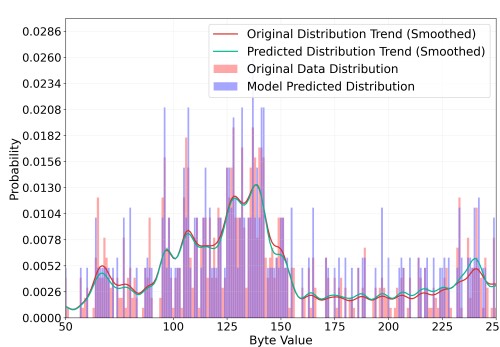

(a) Transition matrix of the Markov chain.          (b) Model's predicted vs. true distributions.

Figure 6: Validation on a synthetic Markovian byte sequence. (a) The transition matrix heatmap shows strong local dependencies, with high probabilities concentrated along the diagonal. (b) A comparison of the true conditional byte distribution (red) and the TimeXer model's predicted distribution (blue).

stabilizes, and the cumulative bpb converges to a steady value. This convergence demonstrates that the models are learning a consistent statistical representation of the data and that our benchmark provides a stable and reliable final score for comparison.

Furthermore, this visualization clearly differentiates the final performance ranking of the models. TimeXer converges to the lowest final bpb of 1.1251, indicating the most effective compression and the best approximation of the data's underlying distribution among the three. It is followed by PatchTST (1.1369) and iTransformer (1.1422). This step-by-step analysis complements the aggregate results in the main paper by illustrating the dynamic behavior of the models and visually confirming their performance hierarchy on the compression task.

### A.12.5 VALIDATION ON SYNTHETIC MARKOVIAN DATA

To provide a definitive validation of our compression-based evaluation paradigm, we designed a controlled experiment using a synthetic dataset whose theoretical properties are perfectly known. We generated a byte sequence from a 256-state Markov chain, where the transition matrix was constructed to exhibit strong temporal dependencies. The probability of transitioning to a new state is inversely proportional to its distance from the current state. This setup creates a data source with a known generative process, allowing us to precisely calculate its theoretical entropy rate. This rate serves as an absolute ground-truth benchmark against which we evaluated our top-performing model, TimeXer, to assess its ability to learn the known data distribution.

The results of this experiment are visualized in Figure 6. The heatmap of the transition matrix in Figure 6 (a) clearly shows this strong local structure, with probabilities heavily concentrated along the diagonal, indicating that the next byte is highly likely to be close in value to the current byte. This is the explicit statistical rule that a successful time series model must learn. Figure 6 (b) demonstrates how well the TimeXer model captured this underlying rule. It compares the true conditional distribution of the next byte against the distribution predicted by the model. The significant overlap between the original and predicted distributions, especially evident in the smoothed trend lines, confirms that the model successfully approximated the data's true generative properties rather than merely memorizing superficial patterns.

The primary advantage of this controlled experiment is the ability to quantify model performance against a perfect theoretical baseline. For the generated sequence with transition probability parameter $p = 0.9$, the theoretical entropy rate was calculated to be 1.268 bits/byte. When evaluated on this data, our top-performing model, TimeXer, achieved an actual compression rate of 1.956 bits/byte. The resulting gap of 0.688 bits/byte provides a direct and unambiguous measure of the model's fidelity in learning the true data distribution. This result strongly substantiates our paper's

Table 10: CR of our best model backbone (TimeXer), specialised time-series compressors, and general-purpose compressors across four datasets.

| Dataset | TimeXer | Sprintz | Elf | Chimp | Camel | Gorilla | LZ4 | Zstd | Brotli | Xz |
|---|---|---|---|---|---|---|---|---|---|---|
| Electricity | **0.1120** | 0.1820 | 0.3065 | 0.3587 | 0.4020 | 0.2269 | 0.3050 | 0.2066 | 0.1969 | 0.1430 |
| ETTh2 | 0.2620 | **0.1220** | 0.8204 | 0.7521 | 0.4790 | 0.7595 | 0.3010 | 0.1506 | 0.1423 | 0.1230 |
| Traffic | **0.1370** | 0.2290 | 0.3120 | 0.8962 | 0.2070 | 0.9794 | 0.3625 | 0.2342 | 0.2226 | 0.1650 |
| Weather | **0.2070** | 0.3160 | 0.4052 | 0.8267 | 0.3570 | 0.7642 | 0.5208 | 0.3372 | 0.3001 | 0.2320 |

central thesis: lossless compression serves as a rigorous, principled, and quantitatively verifiable benchmark for evaluating a model's core ability to capture the underlying generative process of a time series.

### A.12.6 COMPARISON WITH SPECIALIZED COMPRESSORS

To further assess the relative performance of learned models, we compare our best TSCom-Bench backbone against specialised lossless time-series compressors such as Sprintz, ELF, Chimp, Camel and Gorilla, as well as general-purpose compressors (LZ4, Zstd, Brotli, Xz); detailed numbers are reported in Table 10. For clarity, we place the best-performing deep model (TimeXer) in the first column. The results show that TimeXer achieves the lowest or near-lowest compression ratio on all datasets except ETTh2, despite never being designed as a compressor. The weaker performance on ETTh2 is likely due to its limited periodicity and regularity, a well-known characteristic of this benchmark in the time-series literature. Overall, these findings are encouraging: they indicate that modern time-series models already learn distributional structure rich enough to rival specialised compressors. They also suggest that future models explicitly designed for lossless time-series compression may surpass both current deep models and traditional compressors, and that TSCom-Bench provides a natural testbed for exploring this new research direction.

## B COMPARISON WITH CANONICAL TASKS

We provide a detailed comparison between lossless compression and the four canonical evaluation tasks widely used in time series modeling: forecasting, imputation, anomaly detection, and classification.

**Forecasting.** Forecasting aims to predict the future values given the past. The standard loss is mean squared error:

$$\mathcal{L}_{\text{forecast}}(\theta) = \frac{1}{T} \sum_{t=1}^{T} \|x_t - \hat{x}_t^\theta\|_2^2, \quad \hat{x}_t^\theta = \mathbb{E}_{Q_\theta}[x_t \mid x_{<t}]. \tag{37}$$

Minimizing this loss forces $Q_\theta$ to match only the conditional mean. Different distributions can share the same mean but have very different variance or tail behaviour, so a model may achieve low forecasting loss yet diverge from $P$ in KL divergence.

**Imputation.** Imputation requires the model to reconstruct missing values in a partially observed sequence. Let $M \subset \{1, \ldots, T\}$ be a randomly sampled set of masked indices, and let $O$ denote the complement set of observed indices. A typical objective is to minimize the mean squared error on the masked values, denoted by $\mathcal{L}_{\text{imp}}$:

$$\mathcal{L}_{\text{imp}}(\theta) = \mathbb{E}_M \left[ \sum_{t \in M} \left\| x_t - \hat{x}_t^\theta(x_O) \right\|_2^2 \right], \tag{38}$$

where the expectation $\mathbb{E}_M$ is taken over the distribution of masks, and $\hat{x}_t^\theta(x_O)$ is the model's reconstruction of $x_t$ conditioned on the observed values $x_O$. This criterion enforces local accuracy only on masked positions, while unmasked positions are unconstrained. Unless masking covers all possible subsets, $Q_\theta$ can match $\mathcal{L}_{\text{imp}}$ while disagreeing with $P$ elsewhere.

**Anomaly detection.** The model learns the density of normal data and flags deviations. A common approach is to maximize the likelihood on the set of normal data points. Let $T_{\text{normal}} \subset \{1, \ldots, T\}$ be the set of time indices corresponding to normal data. The loss $\mathcal{L}_{\text{anom}}$ is the negative log-likelihood on this subset:

$$\mathcal{L}_{\text{anom}}(\theta) = - \sum_{t \in T_{\text{normal}}} \log_2 Q_\theta(x_t \mid x_{<t}). \tag{39}$$

This objective enforces accurate density estimation only within the restricted support of normal sequences. Probability mass outside this region is largely irrelevant, meaning the model is not penalized for misrepresenting the full distribution.

**Classification.** Classification associates an entire sequence $X$ with a single, discrete label $y \in \mathcal{Y}$, where $\mathcal{Y}$ is the set of all possible labels. The standard objective is to minimize the cross-entropy loss, denoted by $\mathcal{L}_{\text{cls}}$:

$$\mathcal{L}_{\text{cls}}(\theta) = - \log_2 Q_\theta(y \mid X). \tag{40}$$

This objective enforces that the model's conditional label distribution $Q_\theta(y \mid X)$ approximates the true one $P(y \mid X)$, but it does not constrain the sequence distribution $Q_\theta(X)$ itself. A model may achieve perfect classification by exploiting only a few discriminative features, while ignoring most temporal dependencies.

## C  USE OF LARGE LANGUAGE MODELS

During the preparation of this manuscript, we utilized Large Language Models (LLMs), specifically Google's Gemini, as writing assistants. The use of these models was strictly limited to improving grammar, polishing language, and enhancing the clarity of the text. All the core ideas, methodologies, experimental designs, results, and conclusions presented in this paper were conceived and developed exclusively by the human authors. LLMs served solely as a tool for refining the written expression and did not contribute in any form to the scientific content or intellectual contributions of this work.

