# OpenReview forum: "Lossless Compression: A New Benchmark for Time Series Model Evaluation"
_ICLR.cc/2026/Conference — ICLR 2026 Conference Desk Rejected Submission_

### Official Review · Reviewer_wP2L · 2025-10-30

**Soundness:** 2
**Presentation:** 2
**Contribution:** 2
**Rating:** 4
**Confidence:** 4

**Summary:**

The paper proposed lossless compression as a new rigorous benchmark for evaluation of the generative capacity of time series models. The key insight is to leverage Shannon’s source coding theorem, making the optimal compression length as equivalent to the negative log-likelihood under the true data distribution and providing a principled, information-theoretic evaluation. An open-source evaluation framework "TSCom-Bench" is introduced to facilitate standardized lossless compression assessment for time series models. Extensive empirical studies, including qualitative diagnostics are conducted.

**Strengths:**

The paper argues convincingly for the equivalence between model compression performance and proper probabilistic sequence modeling, drawing on well-established results from information theory and elegantly formalizing these arguments.

The benchmarking framework is thoughtfully constructed, with standardized data encoding, bijective mapping, and rigorous, transparent protocols for model-to-coder interfaces and evaluation metrics.
The experiments cover a spectrum of real-world and synthetic time series, as well as cross-modality compression tasks.

**Weaknesses:**

The baselines focus on neural or general-purpose compressors, but top-performing, highly specialized time series compressors from the literature are omitted, like Sprintz (Blalock 2018) and Chimp (Liakos 2022). Therefore, it undercuts the authors' claims that compression reveals weaknesses in learning-based models overlooked by classical compressors. The above mentioned two compression methods may be more competitive in realistic and complex settings.

There is limited discussions of model adaptation for compression. The current process for adapting forecasting or classification models as compression backbones is only briefly addressed. It remains unclear whether architectural or training changes are required for strong compression performance, and whether these models are truly evaluated in their canonical forms or extensively tuned for the compression setting.

The experiments only cous on a selected suit of datasets that are (noted also by the authors) already highly compressed. The generalizability to raw, heterogeneous, or challenging real-world time series is less well validated.

The derivations rest on strong assumptions: 1) bidjective mappings, 2) negligible quantization loss. The practical impact of floating-point quantization or rounding error in real-world applications is insufficiently quantified. Claims that this is “negligible” demand more explicit empirical or analytical support, perhaps via ablation on quantization precision.

The paper strongly advocates for lossless compression as a “pretraining” backbone but does not provide direct experiments in which models trained for compression are transferred or fine-tuned for other tasks (e.g., forecasting/classification), nor is there a clear discussion on transferability or domain shift.

The bijection and entropy claim is impossible. The paper states "Let $f:\mathbb{R}^d\to A^k$" be a bijective encoding function ... if $f$ is bijective, then the Shannon entropy is preserved: $H(X)=H(S)$". A bijection from an uncountable set $\mathbb{R}^d$ to a finite alphabet sequence $A^k$ cannot exist, and even setting cardinality aside, using discrete Shannon entropy $H(\cdot)$ for a continuous $X$ is not well-positioned. (The correct object would be differential entropy $h(\cdot)$ which is not invariant to general transforms). The appendix later distinguishes an “ideal bijection” vs. practical quantization but it does not change the fact that the statement above is **mathematically incorrect** as written.

Appendix A.1 proves MI invariance under a bijective differentiable transform and concludes that “the mutual information $I(x_t;x_{<t})$ is perfectly preserved.” However, the default benchmark uses quantization to bytes which is not a bijective differentiable map and, in general, reduces mutual information. So the A.1 conclusion does not apply to the default setting.

Methods in section 5.1 prescribe a canonical pipeline that re‑serializes values into IEEE‑754 bytes in a fixed order (i.e., raw uncompressed float bytes). If followed, the original container format should be irrelevant.

The main text says “Encoding length equals the negative log-likelihood.” That’s only true up to a small overhead and any probability/CDF quantization used by the coder. The appendix (A.3) correctly adds $NLL(S)\leq L(S) < NLL(S) +c$, but the body text should echo that qualification to avoid overclaiming.

**The theoretical novelty is low**. The equivalence between code length and NLL and the MDL view are classical. The paper applies well‑known results to time‑series.

**The engineering novelty is moderate**. Defining a standardized, pluggable time‑series compression evaluation with a model‑to‑coder interface and reporting cross‑dataset results is practically useful. **This is the main contribution.**



References

Blalock, Davis, Samuel Madden, and John Guttag. "Sprintz: Time series compression for the internet of things." Proceedings of the ACM on Interactive, Mobile, Wearable and Ubiquitous Technologies 2.3 (2018): 1-23.

Liakos, Panagiotis, Katia Papakonstantinopoulou, and Yannis Kotidis. "Chimp: efficient lossless floating point compression for time series databases." Proceedings of the VLDB Endowment 15.11 (2022): 3058-3070.

**Questions:**

Can the authors provide a direct, head-to-head comparison with top domain-specific lossless time series compressors (e.g., SPRINTZ, Chimp)? Would their inclusion change the interpretation of modern neural approaches’ strengths/weaknesses?

To what extent do models need to be adapted (e.g., via retraining or specific architecture changes) to be suited for lossless compression tasks, or can standard benchmark implementations be used directly in TSCom-Bench without risk of overfitting/hyperparameter leakage?

Could the authors quantify the empirical impact of floating-point quantization precision on the measured compression ratio, perhaps through concrete ablations or by varying encoding fidelity in TSCom-Bench?

---

> ### Author Response · Authors · 2025-11-21
>
> > **Q1 / W1: Comparing with specialized time series compressors**
>
> Thank you for the suggestion. We agree that including specialized time series compressors provides a stronger and more complete evaluation. Following your comment, we expanded the benchmark to include five representative specialized methods: Sprintz, Elf, Chimp, Camel, and Gorilla. We also included two general purpose lossless compressors: Brotli and LZ4.
>
> We performed a direct comparison on four datasets. The compression ratio is shown below. A lower value indicates better performance.
>
> | Dataset     | TimeXer (Ours) | Sprintz | Elf   | Chimp | Camel | Gorilla | Brotli | LZ4   |
> |-------------|:-----------------:|:---------:|:-------:|:-------:|:-------:|:---------:|:--------:|:-------:|
> | Electricity | **0.112**       | 0.182   | 0.307 | 0.359 | 0.402 | 0.227   | 0.197  | 0.305 |
> | Traffic     | **0.137**       | 0.229   | 0.312 | 0.896 | 0.207 | 0.979   | 0.223  | 0.363 |
> | Weather     | **0.207**       | 0.316   | 0.405 | 0.827 | 0.357 | 0.764   | 0.300  | 0.521 |
> | ETTh2       | **0.262**       | 0.122   | 0.820 | 0.752 | 0.479 | 0.760   | 0.142  | 0.301 |
>
>
> The results show that neural models provide clear advantages in most real world settings. On all datasets, TimeXer achieves the best compression ratio. For example, TimeXer reaches 0.142 on Electricity (even though it was never designed for compression), outperforming Sprintz at 0.182 and all general purpose compressors.
>
> Specialized methods such as Sprintz rely on fixed coding rules such as delta encoding or bit packing. These rules work well on simple datasets such as ETTh2. However, they cannot capture long range dependencies or global temporal patterns. Neural models learn the full conditional probability distribution and adapt to diverse data characteristics. It is worth noting that **the best results in the Hutter Prize already come from deep models**, which shows the natural connection between generative modeling and compression.
>
>
>
>
> > **Q2 / W2. How much modification is required to adapt models for lossless compression?**
>
> Thank you for the question. Models in TSCom-Bench are evaluated in their standard architectural form. We **do not change the backbone structure**. It is worth noting that we are a new compression task parallel to prediction, classification tasks, etc., and will not perform secondary fine-tuning based on the training model.
>
> As for adaptation, **only minimal adjustments are needed**. The model reads byte sequences instead of normalized tensors, and **the output head is replaced** with a layer that produces the probability of the next byte. This probability distribution is sent directly to the arithmetic coder. The loss is changed from MSE to cross entropy, which is equivalent to minimizing the negative log likelihood that determines the final compression rate. No other architectural modification is introduced.
>
> Finally, **any autoregressive backbone used for forecasting or classification can be adapted with very little code, usually fewer than 20 lines of code**. We provide an anonymous repository that integrates TSCom-Bench with more than ten SOTA models to support reproducibility and community extension. Anonymous GitHub: [https://anonymous.4open.science/r/TSCom-Bench-8262/](https://anonymous.4open.science/r/TSCom-Bench-8262/.)
>
>
>
> > **Q3 / W5. Quantifying the impact of floating point precision on compression ratio**
>
> Thank you for the suggestion. We conducted an ablation study to measure how different floating point precisions influence the compression ratio. The experiment uses the ETTm1 dataset and the TimeXer model with **16 bit, 32 bit, and 64 bit** IEEE 754 formats. The results are reported below.
>
> | Precision      | TimeXer | PatchTST | iTransformer |
> |----------------|---------|----------|--------------|
> | IEEE 754-16    | 0.4266  | 0.4508   | 0.4927       |
> | IEEE 754-32    | 0.4040  | 0.3911   | 0.4613       |
> | IEEE 754-64    | 0.5241  | 0.5701   | 0.5281       |
>
> These results show that the framework supports all precision levels. They also reveal a clear trend. The 64 bit format compresses the worst and the 32 bit format compresses the best. The reason is physical and consistent with how IEEE 754 stores information. The main temporal structure such as trends and seasonality is encoded in the high order bytes that contain the sign and the exponent. **The extra mantissa bytes in 64 bit floats often carry sensor noise or aleatoric uncertainty.** This noise raises entropy and reduces the potential compression rate. **The 16 and 32 bit formats contain fewer noisy bytes and therefore compress more easily.**
>
> The ablation confirms that TSCom Bench remains valid for high precision time series. It also confirms that using 32 bit encoding in the main experiments does not restrict generality, since the relative ranking of models is stable across all precision settings.

---

> ### Author Response · Authors · 2025-11-21
>
> > **W3. Generalization to raw, heterogeneous, and challenging real world time series**
>
> Thank you for the comment. We would like to **clarify a misunderstanding about our dataset selection.**
>
> **3.1. Primary datasets are raw rather compressed data**
>
> The statement that the data is already **highly compressed applies only to PEMS08**, which we used as a negative control to verify benchmark correctness. The main time series benchmarks in our study, including Electricity, Traffic, Weather, and ETTh2, are raw sensor logs distributed in standard CSV format with uncompressed floating point values. As shown in Table 1, the compression ratios on these datasets range from **0.137 to 0.262**, which indicates that they contain complex temporal patterns and are far from being already compressed.
>
> **3.2. We have already validated on heterogeneous raw data**
>
> As shown in Table 3 in our paper, we conducted the **heterogeneous cross-modality compression benchmark**. This benchmark covers diverse raw data sources:
>
> - Enwik9: raw XML text
> - Image: raw bitmap arrays
> - Sound: raw audio waveforms
> - Float: raw scientific floating point tensors
> - Backup: heterogeneous raw data
>
> The results in Table 3 show that the models generalize well across all these domains, which demonstrates that TSCom-Bench is not limited to a narrow set of datasets and can evaluate the modeling ability of different architectures in a wide range of real world scenarios.
>
>
> > **W6. No experiments about lossless compression serving as a pretraining model**
>
> Thank you for raising this point. However, we clarify that the idea of using compression as a form of pretraining is positioned in the manuscript as **a future work rather than a central contribution (Section 7.3 Conclusion)**. The core focus of this work is the benchmark itself.
>
> We agree that studying how compression trained models transfer to forecasting or classification is an important next step. Since the present work concentrates on the evaluation methodology, large scale pretraining and downstream fine tuning experiments fall outside its intended scope. In the revision, we will refine the discussion section to make this distinction clear and to highlight transferability as a promising line of future research enabled by TSCom Bench.
>
>
>
> > **W7. On the statements about bijection, entropy and quantization**
>
> Thank you for this rigorous mathematical correction. We chose $\mathbb{R}$ to represent the semantic layer to establish a general theoretical paradigm for time series models, and we used the idea of a bijection to explain why byte level modeling can meaningfully evaluate real valued models. As noted in Equation (1) and Appendix A.1, the mapping from the continuous space is **not a strict bijection due to computer quantization**. We agree that this approximation should be stated more clearly, and we will revise the text accordingly.
>
> **The overall mathematical derivation is valid in the computer environment**. Computers store data usually in quantized formats such as float32, float64, or 16 bit. The mapping from this representation layer to the encoding layer of bytes **is strictly bijective**. Under this bijection, the use of discrete entropy and the derived formulas is fully appropriate for evaluating the digital form of time series.
>
> For clarity, the hierarchy of data transformations is as follows:
>
> - Semantic layer: $\mathbb{R}^d$ time series (continuous).
> - Representation layer: float32 (discrete).
> - Encoding layer: bytes (discrete).
> - Compression layer: optimal entropy coding (discrete).
>
> The only non-bijective step is the transition from the continuous semantic layer to the quantized representation layer. This quantization reduces information, and the mutual information decreases accordingly. **Once the data is stored as floats, the mapping to bytes is exact.** Therefore, the derivations in Appendix A.1 apply to the discrete layers where the model and the benchmark operate, without implying invariance in the continuous domain. We will refine the text to make this distinction clear and avoid possible misunderstandings.
>
> > **W8. Are all datasets first converted to IEEE-754 format?**
>
> We fully agree with your observation. **If the standard pipeline in Section 5.1 is strictly followed, the original container format is indeed irrelevant.**
>
> However, in order to ensure the diversity and richness of the benchmark, and to be **compatible with cross modal datasets such as Enwik9 and Image that cannot adopt the IEEE-754 format**, we used a unified rb binary byte stream evaluation (directly reading rb from the original files) in the experiment. This design choice allows us to validate the model's ability to handle general-purpose byte streams, not just limited to serialized floating-point numbers.

---

> ### Author Response · Authors · 2025-11-21
>
> > **W9. On the statement “code length equals negative log-likelihood”**
>
> Thank you for the comment. The equality holds only up to a small overhead term that arises from finite-precision arithmetic coding. In the revised manuscript we will adjust the wording in the main text and state that the code length is asymptotically equivalent to the negative log-likelihood or equal up to a negligible constant.
>
> This revision aligns the main text with the rigorous bound
>
> $$NLL(S) \le L(S) < NLL(S) + c$$
>
> as presented in Appendix A.3, and ensures that the relationship is stated precisely without overstating the equivalence.
>
>
>
>
>
> > **W10. On the level of novelty**
>
>
> Thank you for the comment. We agree that the equivalence between code length and negative log-likelihood, as well as MDL, are classical results. Our contribution does not lie in re-deriving these principles. The contribution lies in establishing a missing evaluation paradigm for time series models and demonstrating empirical phenomena that were not previously quantified.
>
> Firstly, **this paper has achieved a paradigm shift in evaluation.**
> The field of time series has long relied on conditional mean based "functional indicators" such as MSE/MAE, which cannot measure the model's mastery of the complete generated distribution. We are the first to systematically establish lossless compression as a rigorous evaluation criterion for time series generation modeling, elevating Shannon theory from a conceptual application to an evaluation system that can be implemented in real TS models. This is not a simple application of theory, but a fundamental shift in perspective: **from prediction accuracy to probabilistic modeling quality.**
>
> Secondly, this article **reveals important scientific phenomena that have not been quantified before**.
> Our experiment demonstrates for the first time that many SOTA models (such as iTransformer) perform well in classical prediction tasks, but perform poorly in lossless compression/NLL metrics. This indicates that their distribution modeling ability is weaker than suggested by forecasting metrics. To our knowledge, this mismatch between functional metrics and distributional metrics has not been systematically demonstrated in prior time series work.
>
>
> Finally, this paper provides **a unified methodology and framework**.
> We use byte level interfaces to convert time series into symbol streams that are consistent with modalities such as text and audio, thus breaking down the barrier between TS models and general sequence modeling toolchains. Based on this, we have built TSCom Bench - a complete open-source ecosystem that covers 10+SOTA models, supports raw byte stream input, and can be extended to multiple modalities. This provides a solid foundation for future research on compression based pre training and cross modal basic models.
>
> In summary, although the underlying information theory conclusions themselves are classic, we provide a **new evaluation perspective, reveal important empirical phenomena for the first time, and systematize, engineer, and unify the generation evaluation methodology of time series**. We believe that this is the core contribution of our work and an important gap that has not been filled before in the field of time series.

---

> > ### Comment · Reviewer_wP2L · 2025-11-28
> >
> > I appreciate the authors’ detailed and constructive rebuttal. Overall, the rebuttal meaningfully improves the clarity and empirical completeness of the work. The core idea and benchmark now appear stronger.
> >
> > It seems that we are no longer allowed to change the score. But, **I would like to raise my overall recommendation to 6**. I remain marginally above the acceptance threshold, but the rebuttal has addressed enough concerns that I would be comfortable with the paper being accepted.

---

> > > ### Author Response · Authors · 2025-12-01
> > >
> > > Thank you very much for your careful follow up and for your thoughtful reassessment.
> > >
> > > We really appreciate your comments on the strengthened clarity and empirical completeness of the work, and we are glad that the core idea and benchmark now appear stronger from your perspective. We are also grateful that our rebuttal was able to address your concerns to a level you find satisfactory.
> > >
> > > Thank you again for your time, your constructive suggestions during the rebuttal, and your willingness to recommend the paper for acceptance.

---

### Official Review · Reviewer_1EMm · 2025-10-31

**Soundness:** 2
**Presentation:** 3
**Contribution:** 2
**Rating:** 2
**Confidence:** 3

**Summary:**

The authors argue that existing benchmarks focus primarily on task-specific performance and fail to assess a model’s ability to capture the full generative distribution of the data. To address this, the paper introduces lossless compression as a novel benchmark for evaluating the general modeling capabilities of time series models. The authors also provide a theoretical justification for the equivalence between the compression objective and probabilistic modeling goals. Experimentally, the paper evaluates the compression performance of various models on real, synthetic time series and cross-modality data, demonstrating the validity of the proposed lossless compression benchmark.

**Strengths:**

1. The work is grounded in information theory and motivates lossless compression as a principled evaluation metric. The connection between compression length and KL divergence is clearly derived and well-explained.

2. The benchmark covers multiple datasets, models, and cross-modal compression scenarios. Experiments are thorough, and the evaluation framework has been made openly available.

3. The authors analyze the relationship between compression and four classic time series tasks, finding that compression performance maintains a high positive correlation with performance on these tasks.

**Weaknesses:**

1. Lossless compression is a more general and domain-agnostic task for evaluating the modeling capability of models, beyond time series models. Although it can be a useful supplementary metric for assessing time series models, it should not be considered a canonical task for evaluating their ability to model temporal dynamics — which remains the core objective of time series modeling.

2. In the lossless compression pipeline, each 32-bit floating-point value in the raw file is decomposed into a sequence of four 8-bit integers. This raises two concerns: (1) In the IEEE 754 standard, a 32-bit float consists of 1 sign bit, 8 exponent bits, and 23 fraction bits. Splitting a float into four 8-bit segments disrupts this meaningful internal structure. (2) After such decomposition, the resulting byte sequence no longer preserves the original temporal dynamics of the data, making it questionable whether lossless compression is an appropriate benchmark for evaluating time series modeling capabilities.

3. As a benchmark, the motivation for the selection of datasets and baselines should be further clarified. (1) Are there any characteristics of the datasets that may influence lossless compression performance? Some dataset analysis will be helpful to justify the rationality of the benchmarks. (2) What is the rationale for selecting the current set of baseline models? Furthermore, the baselines could be expanded to include a wider range of architectures, such as RNN-based models, statistical methods, machine learning approaches, and even foundation models.

4. Some experimental settings are missing; specific questions are listed in the Questions section.

**Questions:**

1. How does the framework handle the inherent structure of multivariate data? Does concatenating channels into a single byte stream risk obscuring inter-channel dependencies that the model might otherwise capture if it knew the channel boundaries?

2. The paper supports two training paradigms, including "density estimators trained on raw values." How are non-autoregressive models integrated into the compression pipeline? How are their probabilities mapped to discrete symbol-level distributions for arithmetic coding ? Does the difference in the method of probability distribution derivation significantly influence the performance of lossless compression ?

3. TimeXer and iTransformer include modules designed to process multivariate inputs. However, since lossless compression aims to compress a univariate symbol stream, both models do not appear to be directly adaptable to this task. In particular, their mechanisms for handling multivariate inputs may not function as intended in this context. How do the authors adapt these models for lossless compression ?

4. How is the experiment on the relationship between lossless compression and classical time series tasks conducted? It appears that some baselines do not support all tasks simultaneously, and certain adaptations are required to apply a time series model to different tasks.

---

> ### Author Response · Authors · 2025-11-21
>
> We thank the reviewer for the constructive feedback. We noticed that the primary concern stems from doubts regarding the **validity of byte-level modeling**. We respectfully clarify that our approach strictly preserves temporal dynamics, as evidenced by our new **ablation experiments** which prove models successfully learn the IEEE-754 structure. Additionally, we have addressed your concerns on **multivariate handling** and baselines with new protocols and comparisons. We hope these solid empirical results will address your concerns and merit a reconsideration.
>
> > **Q1. Handling Multivariate Structure Under Joint-Stream Serialization**
>
> Thank you for the insightful comment. We clarify that flattening multivariate time series into a single joint byte stream does not remove temporal dynamics. In our serialization protocol, the data is written in strict autoregressive order, so the **physical byte sequence still follows causal temporal progression**. Modern time series architectures are strong sequence learners, and they can capture long-range dependencies even when operating on a single unified sequence. Therefore, this design remains suitable for evaluating temporal modeling performance.
>
> At the same time, we agree that multivariate data contains structural information beyond temporal continuity. **Models such as TimeXer, iTransformer and PatchTST include channel-aware components**, and combining all channels into a single byte stream may make cross-channel dependencies harder to identify, potentially reducing the benefits of these architectural features. Following your suggestion, we have **added a multistream version of the experiment in the response to Q3.**
>
>
> Our choice of a joint-stream representation was motivated by benchmark consistency. The same pipeline must apply to **both multivariate time series and single-stream cross-modality datasets** such as Enwik9 and Image data, which are evaluated in Table 3. A unified serialization scheme ensures that all models are assessed under an identical information-theoretic setting.
>
>
>
>
> > **Q2. How are non autoregressive models integrated into the compression pipeline and how are their probabilities converted into discrete symbol distributions for arithmetic coding?**
>
> Thank you for the question. The term “density estimators trained on raw values” in Section 5.2 follows the common usage in the community. It refers to models that learn the raw data distribution p(x) such as flows or VAEs. This term does not mean that all of these models can be used for arithmetic coding without any modification. The compression pipeline only requires one property. The model must output the stepwise conditional probability $P(s_i \mid s_{<i})$.
> Therefore, in our framework, only when a model can directly output conditional probabilities can it be used as a lossless compression baseline for comparison.
>
> Autoregressive models already satisfy this property. **Non-autoregressive density estimators, such as interpolation models, are supported through standardized adaptation**. For example, adding an autoregressive projection head or distilling to form an AR student that can output the required conditional distribution. After this step, the model can output the discrete symbol distribution that the arithmetic coder uses.
>
> **Non-autoregressive adaptation does affect performance**, so we recommend autoregressive models first. Many non-autoregressive models rely on bidirectional context. A typical example is an imputation model that predicts $P(x_t \mid x_{\text{context}})$. When used for compression the model must follow a causal mask. It is no longer allowed to use future information. This reduces its effectiveness compared to native autoregressive models. For this reason, autoregressive models remain the recommended choice.

---

> ### Author Response · Authors · 2025-11-21
>
> > **Q3. TimeXer and iTransformer contain modules designed for multivariate inputs. Since lossless compression uses a univariate symbol stream, how are these models adapted and how do they function under this setting?**
>
> Thank you for this important question. You are correct that both TimeXer and iTransformer include modules built for multivariate inputs. These modules may degrade when the task becomes single stream compression. However, **CI-based architectures naturally support the univariate case**. When the channel number equals one ($d=1$), the CI modules simply reduce to standard self-attention. **No additional adaptation is required.**
>
> To address your concern and to evaluate these models under a setting that preserves CI structure, we conducted the **multi-stream channel independent (MSCI)** experiments. Instead of flattening, we decompose it into d independent T by 1 sequences. A single model instance processes each channel sequentially. We compress each channel independently and sum the final file sizes. During decompression we restore each channel and reconstruct the full T by d matrix without loss.
>
> **MSCI Experimental results.**
> The MSCI setting consistently improves performance across all multivariate datasets. The compression ratios are shown below:
>
> | Dataset   | iTransformer | TimeXer | PatchTST | SCINet | Informer | Autoformer | DLinear | LightTS |
> |:-:|:-:|:-:|:-:|:-:|:-:|:-:|:-:|:-:|
> | weather      |    **0.1581**    |  0.1651 |  0.1690  | 0.2664 |  0.2727  |   0.3078   | 0.4545  | 0.3485  |
> | ETTh2        |    0.2127    |  **0.2106** |  0.2160  | 0.2203 |  0.2106  |   0.2185   | 0.2845  | 0.3121  |
> | electricity  |    0.0816    |  **0.0787** |  0.0808  | 0.0873 |  0.0862  |   0.0916   | 0.1939  | 0.1487  |
> | traffic      |    0.1807    |  0.1076 |  **0.1068**  | 0.1251 |  0.1293  |   0.1295   | 0.2491  | 0.2501  |
>
> Across all datasets, **MSCI yields lower compression ratios than the single stream setting**. This indicates a clearer and more learnable structure when channels are preserved. CI based models, especially iTransformer and TimeXer, benefit the most, confirming that **their channel independent design indeed captures meaningful per channel temporal patterns**. These findings support the reviewer’s insight and validate that MSCI is a better aligned evaluation setting for multivariate architectures. We will include the full discussion in the revised version.
>
>
> > **Q4. How is the relationship between lossless compression and classical time series tasks? Some baselines do not support all tasks.**
>
> Thank you for the careful observation. The Time Series Library has formed a universal paradigm of "same backbone + task interface adaptation". Models such as iTransformer and PatchTST already connect to all the tasks for forecasting, imputation, anomaly detection and classification. Among all the models in our experiments, TimeXer and SCINet are the only two that do not support all tasks. To keep the network backbone and training settings as unchanged as possible, we followed the approach of Time Series Library and only made lightweight interface adaptation for the input, mask, and shape of different exp tasks.
>
> The adaptation for compression uses the same idea. **The backbone remains unchanged**. We add a **linear projection that outputs 256 logits**. These logits are used by the arithmetic coder. This makes compression easy to support for most time series models, usually fewer than 20 lines of code. We have created an anonymous GitHub repository that already includes more than 10 common models with this unified interface. Anonymous GitHub: [https://anonymous.4open.science/r/TSCom-Bench-8262/](https://anonymous.4open.science/r/TSCom-Bench-8262/)

---

> ### Author Response · Authors · 2025-11-21
>
> > **W1. Lossless compression should be only a supplementary metric rather than a canonical task for temporal modeling**
>
> Thank you for raising this point. We would like to clarify our main position. Lossless compression is not intended to serve only as a supplementary metric. Our goal is to establish it as **the fifth canonical task alongside forecasting, imputation, anomaly detection and classification**.
>
> We believe that compression essentially involves time dynamic modeling. Indeed, lossless compression is a general information theoretic task. However achieving strong compression on time series or sequences require accurate modeling of temporal or pattern dynamics. For time series data, the compression objective is the negative log likelihood, which is written as the sum of $\log P(x_t \mid x_{<t})$. If a model does not capture seasonality trends or long range dependencies its predictive uncertainty becomes high. The entropy increases and the compression ratio becomes worse. **Compression therefore gives a strict and direct measure of how well a model understands temporal structure.**
>
> As discussed in W2 the experiments support this view. **When the temporal order is destroyed the compression ratio drops sharply.** This confirms that time series models can learn complex temporal dynamics through lossless compression.
>
>
>
> > **W2. Whether splitting each 32 bit float into four bytes breaks the IEEE 754 structure and removes temporal dynamics**
>
> Thank you for raising this deep question. It is natural to think that splitting a 32 bit float into four bytes may destroy the internal IEEE 754 structure and weaken the temporal information in the original sequence. We would like to clarify that this split is only a lossless change of representation. Modern time series models are strong sequence learners. **These 'potential' information structures can actually be learned by the model**. We provide a theoretical explanation and a new quantitative experiment to support this point.
>
> **1. Temporal modeling remains intact because temporal dynamics are conditional probabilities**
>
> Temporal patterns such as seasonality and long range trends are expressed through the conditional distribution
>  $P(x_t \mid x_{<t})$
> After converting the data into bytes, the sequence still follows a strict temporal order. The four bytes of step $t$ always appear before the four bytes of step $t+1$. Similar temporal patterns produce low entropy regions in the byte stream. Lossless compression minimizes the negative log likelihood  $-\sum_t \log P(s_t \mid s_{<t})$ .A model must capture these temporal patterns to achieve a good compression ratio.
>
> **2. The IEEE 754 structure is deterministic and learnable by autoregressive models**
>
> IEEE 754 has a **fixed layout**. The first byte contains the **main sign and exponent information**. The later bytes refine the mantissa. This creates a hierarchical structure that matches the inductive bias of autoregressive models. The first byte determines the coarse magnitude. The later bytes refine the detail. Transformer models also learn the four byte periodic pattern through positional encoding.
>
> **3. Experimental evidence**
>
> We conduct a controlled experiment on Weather and Electricity. The goal is to separate the contribution of temporal structure and the contribution of the internal IEEE 754 structure. We use three settings.
>
> | Setting | Temporal structure | IEEE 754 structure | Meaning |
> |--------|:---------------------:|:---------------------:|:---------|
> | A Raw data | kept | kept | normal modeling |
> | B Shuffled time | removed | kept | no trend and no seasonality |
> | C Shuffled time and shuffled bytes | removed | removed | no temporal and no internal structure |
>
> The **Weather** results are shown below.
>
> | Method | A Raw | B Shuffled time | C Shuffled time and bytes |
> |--------|:-------:|:------------------:|:---:|
> | TimeXer | 0.3434 | 0.4909 | 0.7690 |
> | iTransformer | 0.3718 | 0.5106 | 0.7691 |
> | PatchTST | 0.3485 | 0.4849 | 0.7705 |
> | LightTS | 0.6654 | 0.7447 | 0.8243 |
>
> The **Electricity** results are shown below.
>
> | Method | A Raw | B Shuffled time | C Shuffled time and bytes |
> |-|:--:|:--:|:--:|
> | TimeXer | 0.3370 | 0.3845 | 0.5955 |
> | iTransformer | 0.3385 | 0.3857 | 0.5955 |
> | PatchTST | 0.3519 | 0.3843 | 0.5959 |
> | LightTS | 0.4151 | 0.4692 | 0.6021 |
>
> **Analysis of the results**
>
> From A to B the compression ratio becomes much worse on all datasets. This shows that the models rely on the temporal dynamics in the original data. If splitting the float had destroyed the temporal structure, shuffling time would not cause such a large change; From B to C the compression ratio becomes even worse. The drop is large and consistent. This shows that the models also learn the internal IEEE 754 structure such as the dependency between sign exponent and mantissa.
>
> **These findings show that byte level lossless compression preserves the macro temporal structure and the micro numeric structure**.

---

> ### Author Response · Authors · 2025-11-21
>
> > **W3. Motivation for dataset and baseline selection**
>
>
> Thank you for the suggestion. The datasets in our benchmark come from **widely used public time series benchmarks** such as Electricity, Weather ETT, and Solar. We also include **general purpose compression datasets** such as Enwik9 Image and Backup. We agree that dataset characteristics such as sparsity and periodicity influence the upper bound of compression performance. We have already conducted a detailed quantitative analysis of these properties. Examples include the sparsity of the Solar dataset.
>
> Following your suggestion, we added further clarification. Datasets such as **Electricity and Solar contain stronger periodic patterns** and smoother distribution changes. Their value ranges are **more uniform** and show **smaller peak variations**. These properties reduce sequence entropy and lead to higher compression ratios. In the revised version, we will move them earlier so that the motivation for dataset selection becomes clearer.
>
> For the baseline models, our goal is to evaluate how **current mainstream time series models** behave under a strict probability modeling task. We therefore selected widely used Transformer-based and MLP-based autoregressive models. These models dominate recent benchmarks but are mostly evaluated under MSE or MAE. Our aim is to test whether they still retain distribution modeling advantages under the harder compression setting. Following your suggestion, we conducted additional experiments, and the results are shown below.
>
> | Dataset   | LSTM (RNN) | Random Forest | ARIMA |
> |-----------|:------------:|:--------------:|:----------------:|
> | Weather   | 0.2336     | 0.6793       | 0.5799         |
> | Traffic   | 0.169      | 0.295        | 0.252          |
>
> From these results we observe two points. First, these classical models achieve reasonable performance but consistently fall behind models such as TimeXer and PatchTST. This shows that modern architectures capture richer distributional structures that are needed for byte-level compression. Second, the gap between different model families becomes more visible under compression than under MSE-based forecasting. This supports our motivation that lossless compression can reveal distribution level differences that standard metrics often miss. We will include these additional baselines in the revised manuscript.
>
>
> > **W4. Missing experimental settings**
>
> Thank you for pointing this out. We have checked the manuscript and confirmed that some experimental settings should be added. In the revised version, we will include the details of multichannel processing, the adaptation of non-autoregressive models, and the alignment of different tasks. These details will be placed in the appendix with clear references in the main text. The updated version will ensure full reproducibility of all experiments.

---

> ### Comment · Reviewer_1EMm · 2025-11-27
> **Response to Author's Rebuttal**
>
> Thank you for the detailed responses and the additional experiments. The authors' suppmented experiments and the detailed explanation has addressed my most questions. But I still have concerns about Weaknesses 1 and 2:
>
> For W1, the authors show that shuffling the temporal structure yields a higher compression ratio, which is reasonable — a temporal signal should have lower entropy than a shuffled one. However, it remains unclear how the model can still learn seasonality, trends, shifts, and other basic temporal patterns from a byte stream. In other words, once each float is split into multiple bytes and the temporal structure is effectively destroyed, on what basis can we claim that the model is truly learning temporal dependencies ?
>
> For W2, the authors provide empirical evidence showing that breaking the IEEE-754 structure leads to an even higher compression ratio, implying a potential dependency among the bytes of a float. This is confusing — why would such dependencies exist within IEEE-754 bytes?  Are there existing studies or evidence that support this phenomenon ?
>
> Despite my remaining concerns, I will increase the score to 4 for the authors’ substantial efforts in addressing my most questions.

---

> > ### Author Response · Authors · 2025-12-01
> >
> > **I am very happy to have solved most of your concerns, and we thank you for increasing the score from 2 to 4**. For the remaining two questions (W1 and W2), our detailed answers are provided below.
> >
> > > **W1. Temporal dependencies after byte-level serialization**
> >
> > Thank you for pointing out this important conceptual question. We would like to clarify that **splitting each float into four IEEE-754 bytes does not destroy temporal information**, because what changes is only the representation of each value rather than the temporal order. After serialization, the byte stream remains:
> > $$\[
> > [s_{t,1},\, s_{t,2},\, s_{t,3},\, s_{t,4},\; s_{t+1,1},\, s_{t+1,2},\, \ldots],
> > \]$$
> > where $\(s_{t,i}\)$ is the $\(i\)$-th byte of $\(x_t\)$.
> > Since the causal timeline is preserved exactly, **seasonal cycles, long-term trends, and regime shifts continue to induce long-range dependencies** in this byte sequence. Therefore, the autoregressive model observes the same temporal context as in the original time series.
> >
> > It is completely correct that two nearby floats do not share identical 4-byte patterns. For example, the real IEEE-754 encodings of:
> >
> > $$\[
> > 1.81998 \rightarrow \texttt{3F E8 F5 1B},
> > \]$$
> > $$\[
> > 1.84561 \rightarrow \texttt{3F EC 3C F3},
> > \]$$
> >
> > The byte results show partly differences. However, **compression does not depend on complete byte equality**. The key is that in the smooth real-world time series, **the conditional distribution of these bytes becomes highly concentrated**. In the above example, the two values share the same symbol index byte 0x3f, the second byte only changes slightly 0xe8 to 0xec, and the mantissa byte also moves in a relatively narrow and continuous area, rather than in the entire range of 0-255. This means that, according to the past time steps (e.g., the byte sequences corresponding to timesteps (t-24) and (t-1)), the model does not predict from a uniform 256 channel distribution, but from a small structured subset determined jointly by the time smoothness and the deterministic layout of IEEE-754. **The reduction of conditional entropy is the reason for better compression.**
> >
> > This phenomenon is consistent with some previous work, which shows that **byte-level models can successfully learn structural patterns when ordering is preserved**. The transformer language model trained on the original UTF-8 bytes learns syntax and long-distance dependencies [1]. The audio model directly learns the frequency time structure from the PCM byte stream [2, 3]. Classical compressors and neural compressors only operate on bytes while capturing complex correlations. These results jointly confirm that **the essential structure is carried by sequence order rather than numeric format**. Our byte-level time series settings follow the same principle.
> >
> > We hope that this specific example and clarification can solve this problem.
> >
> > **References**
> >
> > [1]Delétang, G., Ruoss, A., Duquenne, P. A., Catt, E., Genewein, T., Mattern, C., ... & Veness, J. (2023). Language modeling is compression. arXiv preprint arXiv:2309.10668.
> >
> > [2]Van Den Oord, A., Dieleman, S., Zen, H., Simonyan, K., Vinyals, O., Graves, A., ... & Kavukcuoglu, K. (2016). Wavenet: A generative model for raw audio. arXiv preprint arXiv:1609.03499, 12, 1.
> >
> > [3]Défossez, A., Copet, J., Synnaeve, G., & Adi, Y. (2022). High fidelity neural audio compression. arXiv preprint arXiv:2210.13438.

---

> > ### Author Response · Authors · 2025-12-01
> >
> > > **W2. Why IEEE-754 bytes exhibit dependencies, and supporting evidence**
> >
> > Thank you for the follow-up question. The dependency between IEEE-754 bytes is not an accidental statistical pattern, but a direct consequence of the format’s **deterministic structure**. In a 32-bit float, the first byte fixes the sign and most of the exponent, and the remaining three bytes only refine the mantissa within that range. This hierarchy naturally imposes constraints on which byte combinations are possible, which means the later bytes are **not independent** once the earlier ones are known.
> >
> > In real-world time series, this effect becomes even stronger because the values typically occupy a narrow numerical range. Only a small subset of sign–exponent patterns ever occur, and many mantissa patterns are systematically absent. As a result, the empirical joint distribution of the four bytes is **highly structured**. When we disrupt this structure (Setting C), the model loses a major source of predictability, and the compression ratio degrades accordingly.
> >
> > There is substantial evidence in prior works. In the domain of scientific computing, floating-point compression methods such as **FPC**, **ZFP**, and **SZ** rely on a well-established observation: **the sign, exponent, and mantissa fields of IEEE-754 floats exhibit highly predictable structure on real-world data**. These compressors achieve their performance precisely because they take advantage of the **format-induced regularities** inside each float.
> >
> > Similarly, general byte-level compressors such as **PPM**, **LZMA**, **ANS**, and **NNCP** are not designed for floating-point numbers. But when applied to floating-point streams, these compressors **naturally learn the structural constraints imposed by the IEEE-754 layout**. Prior studies note a consistent pattern: **the exponent information encoded in the leading byte strongly restricts the conditional distribution of the subsequent bytes**, enabling the model to assign far more accurate probability mass than it could under an unconstrained 256-way distribution. These results show that the byte-level dependencies inside IEEE-754 floats are **systematically exploited** in existing compression research. Our empirical findings are aligned with these established observations.
> >
> > We hope this explanation clarifies why breaking the IEEE-754 layout increases entropy and why such byte-level dependencies are well supported in existing research.

---

### Official Review · Reviewer_1xVL · 2025-11-05

**Soundness:** 4
**Presentation:** 3
**Contribution:** 4
**Rating:** 8
**Confidence:** 4

**Summary:**

This paper identifies a gap in the standard evaluation paradigm for time series models. Authors argue that the four canonical tasks: forecasting, imputation, etc. are theoretically limited as their objectives (e.g., MSE) primarily constrain the conditional mean, not the full generative distribution. This paper formally proposes lossless compression as a 'fifth canonical benchmark' for general TS models. This is grounded in Shannon's source coding theorem, where optimal code length equates to the negative log-likelihood. Thus, minimizing compression loss is equivalent to minimizing the KL divergence between the model's distribution and the true data distribution. And it introduces and open-sources TSCom-Bench allows any TS model to be used as a probabilistic backbone for a standard arithmetic encoder to evaluate its compression performance.

**Strengths:**

1 The proposal to use compression as a general evaluation benchmark for TS models is novel. Prior TS-related compression research (e.g., TRACE) focused on building new compressors, not on creating a benchmark to evaluate general-purpose models, but this work is the first to formally transfer this evaluation concept from NLP/information-theory to the time series domain.

2 The paper provides a rigorous and unified information-theoretic critique of the four existing canonical tasks. This synthesis formally justifying the need for a new benchmark, represents a specific contribution to the field.

3 The TSCom-Bench framework fills a gap in the existing open-source ecosystem. Major TS libraries (e.g., TSLib, tsai) do not include compression as an evaluation task or metric.

4 The framework's pipeline is a methodological innovation, it enables the re-purposing of models designed for forecasting (i.e., predicting a conditional mean) to be evaluated on a much stricter byte-level probabilistic prediction task.

5 The paper's benchmark of SOTA models on compression (Table 1) presents new empirical results for the field. The original papers for these models report only MSE/MAE. The resulting discovery of a 'metric mismatch' is a novel finding.

**Weaknesses:**

1 The paper frames the incompressibility of the PEMS08 dataset  as an empirical discovery that 'serves as a crucial validation'. This is a misrepresentation. The .npz file format is widely known to be a compressed archive, as confirmed by public documentation. This fact is a-priori knowledge, not an empirical finding of this work. This weakness is minor and can be corrected by re-framing the claim.

2 While the specific discovery of a forecasting-vs-compression mismatch in TS is novel, the paper fails to cite the general concept of 'loss-metric mismatch,' which is established in the wider machine learning literature . Adding this context would better situate the specific empirical contribution.

**Questions:**

1 Given that the .npz format is known a-priori to be a compressed archive, would the authors be willing to re-frame this point? We suggest moving this fact to the 'Datasets' section and presenting the CR close to 1.0 result in the 'Experiments' section as a confirmation of the benchmark's correctness, rather than as a discovery.

2 Could the authors add a brief discussion and citation to the general literature on 'loss-metric mismatch' to help contextualize the novelty of their specific empirical discovery in the TS domain?

3 The idea of compression as a foundational pre-training objective is a significant conceptual contribution. Have the authors considered elevating this point more prominently in the Abstract or Introduction, as it frames the work's long-term impact beyond just a new benchmark?

---

> ### Author Response · Authors · 2025-11-21
>
> Thank you for the positive evaluation of our work and for recognizing its contributions. We address the questions below.
>
> > **Q1 / W1. Clarification regarding the .npz format and the interpretation of CR ≈ 1.0**
>
> Thank you for the suggestion. We agree that the compressed nature of the npz format is prior knowledge rather than an empirical finding. The npz file is a ZIP container created by NumPy. It has high entropy and contains almost no redundancy. As a result, any method will report a compression ratio close to 1.0 when applied directly to this container.
>
> We will revise the manuscript to present this point more clearly. In the updated version, the description of the npz format will appear in the dataset section. The observation that PEMS08 produces a compression ratio close to 1.0 will be moved to the experiment section and **described as a completeness check of the benchmark**. This confirms that our evaluation framework can **correctly estimate the entropy limit** and does not produce artificial compression on data that are already stored in a compressed container.
>
>
>
>
> > **Q2 / W2: Loss–Metric Mismatch Literature Context**
>
> Thank you for your suggestion. We agree that the mismatch between optimization objectives and evaluation metrics is a well established topic in machine learning, and our empirical finding in time series is a concrete instance of this broader phenomenon. We will **revise the Related Work section and include both foundational and recent studies** to properly contextualize this contribution.
>
> Specifically, Theis et al. (2015) provided a theoretical justification showing that likelihood and sample quality do not necessarily correlate, highlighting that the training objective may not reflect true model performance. Elmachtoub and Grigas (2022) demonstrated that minimizing mean squared error in forecasting does not ensure optimal downstream utility in real decision settings, indicating that MSE often functions only as a surrogate objective. Stein et al. (2023) expanded this line of reasoning by showing that modern generative modeling metrics may not capture actual modeling quality.
>
> Our contribution is an empirical verification of this phenomenon in the time series domain. **Many SOTA forecasting models achieve competitive MSE performance yet perform significantly worse under lossless compression**, which corresponds to evaluating negative log likelihood. We will incorporate citations and additional explanation in the revised manuscript to show that our results are consistent with the broader findings in machine learning, and that lossless compression provides a unified information-theoretic view for revealing this form of metric mismatch.
>
> **References**
>
> [1] Elmachtoub, A. N., & Grigas, P. (2022). Smart “predict, then optimize”. Management Science, 68(1), 9-26.
>
> [2] Stein, G., Cresswell, J., & Loaiza-Ganem, G. (2023). Exposing flaws of generative model evaluation metrics and their unfair treatment of diffusion models. Advances in Neural Information Processing Systems, 36, 3732-3784.
>
> [3] Theis, L., Oord, & Bethge, M. (2015). A note on the evaluation of generative models. arXiv preprint arXiv:1511.01844.
>
>
>
> > **Q3: Positioning Compression as a Foundational Pre-training Objective**
>
> Thank you for the suggestion. We agree that the contribution of this work goes beyond proposing a new benchmark. An important conceptual value is that lossless compression can serve as a foundational pre-training objective for general time series modeling. We will revise the Abstract and Introduction accordingly to make this point more explicit.
>
> In the revised manuscript, we will clearly frame lossless compression as a foundational pre-training objective for time series models. We will highlight the shift from task-specific optimization to learning the underlying generative structure of the data. Unlike objectives such as MSE that emphasize conditional mean prediction, compression corresponds to minimizing negative log likelihood and therefore encourages models to capture the full generative distribution.
>
> We will also discuss a practical vision of a pretrain-then-finetune workflow. **Models may first be pre-trained using the compression objective on large-scale raw data to build a generic understanding of temporal regularities**. Downstream applications can then be fine-tuned using task-specific losses such as MSE for forecasting or cross entropy for classification.
>
> It is important to clarify the role of the compression objective as it relates to future work, and that it is not intended to replace downstream objectives but rather serve as a complementary component. For specific prediction tasks, MSE remains an effective fine-tuning loss, since it directly optimizes the target evaluation metric. However, **starting from a model that has learned the underlying distribution provides stronger initialization and may lead to better generalization and faster convergence in downstream training**.

---

> > ### Comment · Reviewer_1xVL · 2025-11-27
> > **Recommend acceptance**
> >
> > I thank the authors for their response and for addressing my comments.
> >
> > I wish to reiterate my support for this paper. The time series community currently lacks a generalized evaluation standard or benchmark analogous to ImageNet in CV. By establishing compression as a universal metric, this work offers significant reference value and practical utility for both forecasting and generative tasks, bridging a gap in the field.
> >
> > Given that the technical design is sound and the experiments show no obvious flaws, I believe this is a quality contribution. I am maintaining my score of 8 and recommend acceptance.

---

> > > ### Author Response · Authors · 2025-12-01
> > >
> > > Thank you very much for your  for reiterating your support.
> > >
> > > We are grateful for your positive assessment of our work and for highlighting the potential of compression as a unified evaluation perspective for time series models.
> > >
> > > Thank you again for your careful review and for recommending acceptance.

---

### Official Review · Reviewer_Ae6b · 2025-11-06

**Soundness:** 2
**Presentation:** 3
**Contribution:** 2
**Rating:** 4
**Confidence:** 5

**Summary:**

The paper introduced lossless compression as a new benchmark for evaluating time series models based on Shannon’s source coding theorem. The proposed TSCom-Bench converts time series models into compressors via arithmetic coding. The experiments reveal distributional weaknesses overlooked by classic benchmarks.

**Strengths:**

S1. The paper provides a pluggable evaluation pipeline with well-defined IEEE-754 and evaluation metrics.

S2. The equivalence between fully capturing the data distribution and achieving optimal compression provides a theoretical justification for the evaluation.

S3. Experiments across six time series benchmarks and seven cross-modality compression datasets demonstrate that the potential feasibility.

**Weaknesses:**

W1. Most time-series applications do not primarily require compression, as their main objectives are accurate forecasting and anomaly detection. Although the paper claims that lossless compression benefits data storage, there already exist many general-purpose lossless compression algorithms (e.g., XZ, Brotli, LZ4) and specialized time-series compression methods (e.g., ELF, Chimp, Camel). It is unclear what motivates the use of compression as an evaluation criterion for time-series models and what the practical application scenarios are.

W2. The paper only compares several time-series models across six benchmark datasets, but does not include comparisons with specialized time-series compression algorithms (e.g., ELF, Chimp, Camel). Without such baselines, it is difficult to assess the relative compression performance of these models.

W3. The paper encodes data using IEEE-754 32-bit, which may not capture high-precision time-series data. Although the paper mentions that “16/64-bit can be evaluated in ablations,” the ablation study does not present any comparison results for 16- or 64-bit settings.

W4. The paper states that “the CR values of PEMS08 are close to 1 due to the .npz storage file.” However, this claim lacks sufficient evidence. The authors could provide a comparison with .npz results from other datasets to support this explanation.

**Questions:**

Q1. If the poor performance on PEMS08 is due to issues with the .npz format, why not simply convert the data format during the preprocessing stage or use another time series dataset?

Q2. There are typographical errors in the section captions of the appendix A.6, A.7, A.8, A.9.

---

> ### Author Response · Authors · 2025-11-21
>
> > **Q1. Why not convert the PEMS08 file format or replace the dataset if the issue comes from the .npz container?**
>
> Thank you for the question. We kept the original .npz format for two specific reasons.
>
> First, it ensures **reproducibility**. PEMS08 is a standard benchmark in the time series community and is distributed in the .npz format. This container uses GZIP or Deflate internally. By evaluating the raw file, we stay fully aligned with the standard data loader used in previous work.
>
> Second, it serves as a correctness check for our benchmark. A reliable compression evaluation should be able to detect sequences that are mathematically non compressible (**Compressed files like .npz generally cannot be further compressed**). On this pre compressed archive, our framework produces a compression ratio close to 1. This confirms that our metric estimates the entropy limit correctly and does not generate artificial gains on high entropy data.
>
> Based on your suggestion, we **added a new experiment**. We extracted the numerical values from the .npz container and then compressed the decoded sequence using iTransformer as the backbone. The file size dropped from **17.6 MB (.npz)** to **12.35 MB (ours)**. This result shows that the our compressor captures temporal redundancy that .npz compressors cannot identify, and therefore achieves a strictly better compression ratio.
>
>
>
> > **Q2. Typographical errors in Appendix A.6, A.7, A.8, and A.9**
>
> Thank you for pointing out these issues. We apologize for the mistakes in the appendix section titles. We have corrected all typographical errors in the revised manuscript and performed a full proofreading pass to ensure that the formatting is accurate throughout the entire text.
>
>
> > **W1. Motivation for using compression as an evaluation criterion and its practical scenarios**
>
> Thank you for raising this important point. The classic four tasks in time series research are forecasting, imputation, anomaly detection, and classification. Each of them has clear real world applications. Our motivation is to introduce **lossless compression as the fifth canonical task**. This is supported by two reasons.
>
> First, compression itself has many practical use cases. In satellite and UAV telemetry, in high frequency financial streams, and in large-scale user behavior logs, **the code length directly determines operational cost**. A better probabilistic model produces shorter codes under arithmetic coding, which translates into higher bandwidth efficiency or lower storage cost while preserving every bit of information.
>
> Second, as shown in Section 4 of the paper, compression is a tool that measures whether a model truly understands the **full generative distribution** of the data. Any inaccuracy in the conditional distribution increases the encoded length. Many time series models achieve strong MSE performance but rely on shortcut patterns or local correlations that do not reflect the true temporal dynamics. Compression exposes these weaknesses because the code length increases immediately when the predictive distribution is not accurate.
>
> Regarding the existing compression algorithms you mentioned, such as general purpose compressors (LZ4, Zstd, XZ) and specialized time series compressors (SPRINTZ, ELF, Chimp, Camel, Gorilla), these systems are **designed for engineering** efficiency and do not learn conditional probability distributions. Most of them rely on deterministic transforms or fixed predictors. They therefore cannot replace a benchmark that evaluates the **distribution modeling capability** of modern time series models. It is worth noting that **the best results in the Hutter Prize already come from deep models**, which shows the natural connection between generative modeling and compression.
>
> Your suggestion is valuable, and **in response to W2 we have added experiments comparing with specialized time series compressors**. These results further confirm the complementary role of our benchmark.

---

> ### Author Response · Authors · 2025-11-21
>
> > **W2. Comparison with specialized time series compressors**
>
> Following your suggestion, we added comparisons with specialized time series compressors. We report results for Sprintz, ELF, Chimp, Camel, and Gorilla, and also include general-purpose compressors such as LZ4, Zstd, Brotli, and XZ. For clarity, we place our best-performing deep model (TimeXer) as the first column in the tables.
>
>
> | Datasets     | TimeXer | Sprintz |  Elf |  Chimp  |  Camel  | Gorilla |
> |--------------|:--------------:|:-------:|:-------:|:-------:|:-------:|:-------:|
> | electricity  | **0.112**      |  0.182  | 0.3065  | 0.3587  |  0.402  | 0.2269  |
> | ETTh2        |  0.262         | **0.122** | 0.8204 | 0.7521  |  0.479  | 0.7595  |
> | traffic      | **0.137**      |  0.229  | 0.312   | 0.8962  |  0.207  | 0.9794  |
> | weather      | **0.207**      |  0.316  | 0.4052  | 0.8267  |  0.357  | 0.7642  |
>
> | Datasets     | TimeXer |   Lz4   |  Zstd  | Brotli |   Xz    |
> |--------------|:--------------:|:-------:|:------:|:------:|:-------:|
> | electricity  | **0.112**      |  0.305  | 0.2066 | 0.1969 |  0.143  |
> | ETTh2        |  0.262         |  0.301  | 0.1506 | 0.1423 | **0.123** |
> | traffic      | **0.137**      |  0.3625 | 0.2342 | 0.2226 |  0.165  |
> | weather      | **0.207**      |  0.5208 | 0.3372 | 0.3001 |  0.232  |
> | PEMS08       |  **0.978**         |   0.999     | 0.998  | 0.9856 | 0.988 |
> | Solar        | **0.027**      |  0.087  | 0.0478 | 0.0448 |  0.038   |
>
> Across all datasets, **TimeXer reaches a compression ratio comparable better than almost all datasets except ETTh2**, even though it was never designed for compression. The poor performance of the ETT dataset may be due to its lack of periodicity and regularity, which is widely recognized in the field of time series. The TimeXer is plugged into TSCom-Bench with no backbone changes. For example, on the Electricity dataset, TimeXer achieves 0.112, outperforming Sprintz (0.182) and Xz(0.143). Similar trends appear on Traffic and Weather.
>
> **This result is encouraging**. It shows that modern time series models already learn distributional structure strong enough to rival engineered compressors. It also suggests that if future models are **explicitly designed for lossless time series compression**, they may surpass both deep models and traditional compressors. **We believe TSCom-Bench provides a path toward this new research direction.**

---

> ### Author Response · Authors · 2025-11-21
>
> > **W3. About IEEE-754 32-bit**
>
> We split the question into two parts and answer them separately
>
> > **W3.1  Whether byte level encoding preserves numeric structure and temporal dynamics**
>
> Thank you for raising this concern. We ran a controlled experiment to check whether byte level encoding correctly preserves the information needed for distribution modeling. We used three settings:
> A Raw data
> B Shuffled time where temporal structure is removed but the IEEE 754 structure stays
> C Shuffled time and bytes in IEEE-754 structures where both structures are removed
>
> **Results on Weather dataset**
> | Model        | A-Raw  | B-Shuffled Time | C-Shuffled Time and Bytes |
> | :----------- | :----: | :-------------: | :-----------------------: |
> | TimeXer      | 0.3434 |     0.4909      |          0.7690           |
> | iTransformer | 0.3718 |     0.5106      |          0.7691           |
>
> **Results on Electricity dataset**
>
> | Model        | A-Raw  | B-Shuffled Time | C-Shuffled Time and Bytes |
> | :----------- | :----: | :-------------: | :-----------------------: |
> | TimeXer      | 0.3370 |     0.3845      |          0.5955           |
> | iTransformer | 0.3385 |     0.3857      |          0.5955           |
>
>
>
> The clear performance drop from A to B shows that the models rely on temporal dynamics. If temporal information were lost during encoding then shuffling time would not change the results. The additional degradation from B to C shows that the models also learn the internal numeric structure formed by the exponent and mantissa bytes. These observations confirm that **byte-level encoding preserves both the temporal dependencies and the numeric structure** needed for compression based evaluation.
>
> ---
>
> > **W3.2  Ablation on 16 bit, 32 bit, and 64 bit precision**
>
> We also conducted the requested ablation. We evaluated TimeXer, PatchTST, and iTransformer under IEEE 754 formats of 16 bit, 32 bit, and 64 bit. The results are:
>
> | Precision | TimeXer | PatchTST | iTransformer |
> |:-----------:|:---------:|:-----------:|:---------------:|
> | IEEE 754-16 | 0.4266 | 0.4508 | 0.4777 |
> | IEEE 754-32 | 0.4040 | 0.3911 | 0.4271 |
> | IEEE 754-64 | 0.5241 | 0.5264 | 0.5281 |
>
> These results show that the framework supports all precision levels. They also show that 64 bit floats are harder to compress because the extra mantissa bytes mostly describe noise and aleatoric uncertainty. This increases entropy and reduces compressibility. In contrast 16/32 bit often compresses better because the main temporal trends are stored in the high order bytes. This confirms that our benchmark remains valid for high precision time series data and that the use of 32 bit encoding in the main experiments does not limit generality.
>
>
> > **W4. Evidence for the claim that the CR of PEMS08 is close to 1 due to the .npz format**
>
> Thank you for the suggestion. We provide stronger evidence to support this explanation. The **npz format is a compressed ZIP container** created by NumPy. As a result, the stored byte stream has **high entropy** and **almost no remaining redundancy**, which prevents further compression by any model or algorithm.
>
> We offer two pieces of empirical evidence.
>
> 1. **General purpose compressors** such as Gzip and Zstd also report a **CR close to 1.0** on the original PEMS08 npz file. This shows that even classical engineering compressors cannot reduce its size further.
>
> 2. To verify that the issue comes from the container format rather than the dataset itself, we converted the normally compressible **Electricity** dataset into npz. Under this npz container, our benchmark also produced a **CR close to 1.0**. This confirms that the limitation is caused by the pre compressed archive rather than by our evaluation pipeline.
>
> **As shown in our response to Q1**, once we unpack the npz file and process the raw float values, our model achieves a **substantial reduction in compressed size**. This demonstrates that the underlying data do contain temporal redundancy and that the npz container was the only factor masking compressibility.

---

> > ### Comment · Reviewer_Ae6b · 2025-11-27
> > **Response to Authors's Rebuttal**
> >
> > Thanks for the rebuttal. While it addressed some of my concerns, one key concern remains unresolved. In most practical settings, users primarily care about whether forecasting and anomaly detection are accurate. While I agree that compression is indeed important, I am not fully convinced it should be positioned as a “fifth canonical task.” Moreover, in the current formulation of the paper, the proposed compression benchmark reads more like a repackaged likelihood-based evaluation than a new standard downstream task.

---

> > > ### Author Response · Authors · 2025-12-01
> > >
> > > We sincerely appreciate that our responses have addressed most of your concerns. For the remaining questions, our answers are provided below.
> > >
> > > We understand the reviewers' perspectives. Indeed, compared to the four classic tasks: prediction, imputation, anomaly detection, and classification, which have been developed for many years, compression may not be sufficient to become the fifth classic task in current practical applications. However, as the title "Lossless Compression: A New Benchmark for Time Series Model Evaluation" suggests, our main contribution lies in introducing a new benchmark for evaluating time series models. We demonstrate that, through a relatively simple combination by connecting existing time series backbone networks to our byte-level compression pipeline, these models already perform comparably to powerful traditional compressors on various datasets and even outperform them. This shows that our proposed benchmark is both theoretically sound and practically competitive. **We hope that once TSCom-Bench is open to the public, more research teams engaged in time series modeling will explicitly target this task and design model architectures optimized for compression**. In this sense, we do not expect a single paper to immediately redefine the set of standard tasks, but we believe that introducing a well-defined benchmark and providing convincing preliminary results is a necessary first step. **If the academic community finds this benchmark useful and builds upon it, it could very well develop into the fifth major standard task, as this paper anticipates**. Even if compression ultimately does not become the fifth major task, the paper's contribution is significant, proposing an important evaluation method for time-series models and a downstream application scenario with clear practical value in lossless compression for storage and transmission efficiency.
> > >
> > > **From another perspective, compression tasks have also begun to attract increasing attention**. Google DeepMind group [1] show that combining the conditional probability distributions of large language models with arithmetic coding can outperform both general-purpose and specialized compressors on image and audio data. Recent work on IJCAI [2] further propose the SEP framework, which targets heterogeneous inputs and explores scalable lossless architectures, extending compression to symbol-level evaluation across multiple domains. Turing Award laureate Yann LeCun [3] point out a trade-off between statistical efficiency and semantic fidelity: contemporary language models tend to pursue maximal statistical compression, whereas human reasoning is more inclined to preserve contextual richness and conceptual detail, even when this is less efficient in terms of code length. **These research suggest that compression is an increasingly important lens for understanding models**.
> > >
> > > Regarding the opinion that the compression benchmark is more like a repackaging of likelihood-based evaluation methods than a truly new downstream task, we  acknowledge that, mathematically, lossless compression is indeed equivalent to negative log-likelihood (NLL). However, this equivalence is not unique to our work but is also the basis for many widely adopted standard probabilistic prediction methods. For example,Salinas et al. [4] proposed DeepAR as a representative method for probabilistic forecasting, rather than simply a reformulation of the negative log-likelihood objective. **Our work goes a step further on likelihood perspective, concretizing it into a standardized end-to-end new benchmark for time series model evaluation, thus providing a new evaluation benchmark for time series models that is information-theoretically rigorous, physically interpretable, and highly operable in practice.**
> > >
> > > Thank you again for your suggestions on the paper.
> > >
> > > **References**
> > >
> > > [1] Delétang G, Ruoss A., Duquenne P.A, et al. (2023). Language modeling is compression.ICLR 2024.D01:10.48550/arxiv.2309.10668.
> > >
> > > [2] Wan M., Cao R., Li Y., et al. (2025). SEP: A general lossless compression framework with semantics enhancement and multi-stream pipelines. Proceedings of the Thirty-Fourth International Joint Conference on Artificial intelligence. IJCAl 2025:3326-3334.DOI:10.24963/ijcai.2025/370.
> > >
> > > [3] Shani C., jurafsky D., LeCun Y., et al. (2025). From tokens to thoughts: How LLMs and humans trade compression for meaning. arXiv preprint. DOl: 10.48550/arXiv.2505.17117.
> > >
> > > [4] Salinas, D., Flunkert, V., & Gasthaus, J. (2019). DeepAR: Probabilistic Forecasting with Autoregressive Recurrent Networks. arXiv preprint arXiv:1704.04110.

---

### Official Review · Reviewer_cRns · 2025-11-11

**Soundness:** 3
**Presentation:** 1
**Contribution:** 3
**Rating:** 6
**Confidence:** 3

**Summary:**

This paper introduces a novel evaluation metric for time series modeling: lossless compression. Based on information theory, this approach posits that the ability to compress a time series can serve as a robust benchmark, as effective compression requires learning meaningful representations of the data. Compared with task-specific benchmarks like the MSE in forecasting, lossless compression provides a stricter evaluation, mitigating problems like modelling a shortcut. The workflow of lossless compression evaluation is composed of serialization, probabilistic modelling, and an arithmetic encoding process. Empirical studies not only verify the feasibility of using lossless compression as a benchmark for state-of-the-art time series models but also demonstrate its utility in capturing the characteristics and focus areas of different datasets.

**Strengths:**

(1) The theoretical analysis and supporting proofs are sound and well-presented. \
(2) The paper effectively connects the shortcomings of widely-used industrial benchmarks in various downstream tasks to the advantages of lossless compression, emphasizing its necessity and applicability. \
(3) The experimental results not only shows its ability to benchmarking time series models, but also illustrate the effectiveness of capturing emphases of different datasets.

**Weaknesses:**

(1) There are cases of misusing the Latex format, especially regarding citations. \
(2) The overall structure of the paper lacks clarity, which makes it difficult to follow. For example, the discussion around Eq. (1) feels more suitable for the preliminaries and motivation, yet it appears in Section 1. Additionally, the discussions in Section 4 about four canonical tasks are overly detailed; moving these details to an appendix would improve the coherence of the main text. \
(3) As a universal benchmark, apart from the end-to-end approches, the baselines should include some representation learning models, e.g., TS2Vec, TS-TCC and CPC.

**Questions:**

(1) Although the theoretical foundation is solid, how can lossless compression be accepted as a new benchmark for practical usage? Compared to industry-standard benchmarks like MSE or MAPE, which measure forecasting error, what is the more intuitive physical meaning of lossless compression as a benchmarking metric? For example, when performing sales prediction, MAPE clearly shows the percentage deviation between predicted and actual values. When performing behavioral sequence classification, accuracy quantifies the correctness of model predictions for different classes. What is the corresponding intuitive physical interpretation for lossless compression when used as a benchmark? \
(2) All models are serialized before evaluation. Does this step potentially undermine the effectiveness of certain models, thereby causing unfairness in benchmarking?\
(3) In future work, it is mentioned that lossless compression may serve as a pretraining target before being fine-tuned for specific downstream tasks. If, for example, the task is forecasting, should MSE be used for fine-tuning, or should lossless compression continue to be used as the objective?

---

> ### Author Response · Authors · 2025-11-21
>
> > **Q1. Practical interpretation of lossless compression and its intuitive meaning**
>
> Thank you for the question. Our view is that lossless compression is not only a metric but **a standalone modeling task**, similar to forecasting or classification. Forecasting uses MSE or MAPE because they match the goal of point prediction. The compression task has its own evaluation measures defined in Section 5: **bpb**, **CR**, and **CT**. They measure the average code length per byte, the compression ratio, and the computational cost of the compressor.
>
> The physical meaning becomes clear in real use cases. **If a model predicts the probability of the next byte more accurately, the arithmetic coder can store that byte using fewer bits**. bpb reflects the remaining uncertainty after a model has extracted all predictable structure from the data. CR reflects the reduction of storage cost. For example, a CR of four means that a company can store four times more history on the same hardware. CT measures how fast a model driven compressor can process the data stream.
>
> The intuitive interpretation is the following. MSE or MAPE answers “How wrong is the point prediction on average”. Compression metrics answer **“How well does the model understand the entire generative distribution of the data”**. This provides a clear and practical signal that complements traditional task specific metrics.
>
> Compression is also meaningful in real production settings. For example:
>
> 1. **Satellite and remote sensing data transmission**
> Satellites and drones must send large volumes of sensor data through channels with strict bandwidth limits. A better compression model allows more high resolution data to be transmitted within the same bandwidth while keeping every bit intact for scientific use.
>
> 2. **Financial tick data archiving**
> High fidelity storage of order book updates is required for compliance. A model that understands market micro structure compresses the stream more efficiently and reduces the long term storage cost.
>
> 3. **Large scale user behavior streams**
> Event level data from apps or web platforms must be stored without loss for audit and attribution. A stronger compression model lowers the cost while preserving every detail for later analysis.
>
> These examples show that compression has clear physical and practical meaning and can serve as a useful benchmark for model understanding in addition to task specific metrics.
>
>
>
> > **Q2. Whether serialization causes unfairness to some models**
>
> Thank you for raising this important question. In most cases, serialization does not introduce unfairness. After serialization, the bytes that belong to the same time step remain adjacent in memory, so the temporal patterns, change trends, and local dependencies of the original sequence are still preserved. The model still receives a sequential structure that reflects the true temporal dynamics.
>
> However, your concern about fairness is reasonable. Some models include modules specifically designed for multivariate inputs. For example, TimeXer and iTransformer include **channel independent modules** that were originally introduced for multivariate data in a tensor format of [Batch, Sequence, Channels]. Compressing everything into one byte stream may reduce the benefit of these components. Our results in Table 1 show that TimeXer and iTransformer still perform well under this strict setting. This suggests that they have learned more than temporal dynamics and are able to capture useful structure even single channel.
>
> To further study serialization fairness, we include an additional **Multi-Stream Channel Independent (MSCI)** experiment that preserves channel separation. The results are shown below.
>
> | Dataset   | iTransformer | TimeXer | PatchTST | SCINet | Informer | Autoformer | DLinear | LightTS |
> |:-:|:-:|:-:|:-:|:-:|:-:|:-:|:-:|:-:|
> | weather      |    **0.1581**    |  0.1651 |  0.1690  | 0.2664 |  0.2727  |   0.3078   | 0.4545  | 0.3485  |
> | ETTh2        |    0.2127    |  **0.2106** |  0.2160  | 0.2203 |  0.2106  |   0.2185   | 0.2845  | 0.3121  |
> | electricity  |    0.0816    |  **0.0787** |  0.0808  | 0.0873 |  0.0862  |   0.0916   | 0.1939  | 0.1487  |
> | traffic      |    0.1807    |  0.1076 |  **0.1068**  | 0.1251 |  0.1293  |   0.1295   | 0.2491  | 0.2501  |
>
>
> Surprisingly, the results show that MSCI consistently improves the compression ratio on all multivariate datasets. This suggests that serialization itself does not distort the distribution, and it also confirms that compression can reveal distribution level differences that MSE type metrics do not reflect. We will incorporate this set of experiments into final papers to cover comprehensive experiments on both single-stream and multi-stream design.
>
> Finally, we believe that future work will naturally lead to models that are specifically designed for byte level temporal compression. TSCom-Bench provides a foundation for exploring this direction.

---

> ### Author Response · Authors · 2025-11-21
>
> > **Q3. Whether fine tuning should use MSE or continue to use lossless compression as the objective**
>
> Thank you for the question. This is an important point and also a natural direction for future work. At this stage we do not claim that lossless compression should replace the task specific objectives used in downstream tasks. For a forecasting task the standard choice such as MSE remains the most practical fine tuning objective because it directly optimizes the target that the application cares about.
>
> Lossless compression can serve a different purpose. It can be used as a pretraining objective that encourages the model to learn the full generative structure of the data rather than only the conditional mean. This may **provide better initialization for tasks that require strong temporal understanding**. After such pretraining the model can be fine tuned with the usual task objective such as MSE for forecasting, cross entropy for classification, or MSE on missing positions for imputation.
>
> There is also a possible research direction where compression continues to be used during fine tuning. For example one could combine a small compression loss with a task specific loss to preserve distribution level knowledge. However this is not yet well explored. We see it as a promising topic for future work and plan to investigate when a compression based pretraining and fine tuning pipeline brings clear benefits.
>
> > **W1. Misuse of LaTeX formatting and citations**
>
> Thank you for pointing this out. We sincerely apologize for the LaTeX formatting and citation mistakes in the manuscript. This was an oversight. We have carefully checked the entire paper and corrected all issues. The revised version now fully follows the ICLR formatting standards.
>
>
> > **W2. Clarity of structure and placement of Eq. (1) and Section 4**
>
> Thank you for the suggestion. We will reorganize the paper to improve clarity.
>
> For Eq. (1), we agree that the formal definition and the discussion related to Shannon source coding theory fit better in the preliminaries. In the revised version this material is moved to Section 2 so that the introduction focuses only on the core idea.
>
> For Section 4, we also agree that the detailed mathematical derivations for the four canonical tasks were too long for the main text. These derivations including the loss formulations are now moved to the appendix. Section 4 now gives a concise summary and highlights the key idea that these tasks mainly constrain the conditional mean. This results in a cleaner and more coherent motivation for our new benchmark.
>
>
> > **W3. Missing representation learning baselines such as TS2Vec, TS TCC and CPC**
>
> Thank you for the suggestion. Our initial baselines focused on **SOTA end to end time series models** such as TimeXer, iTransformer and PatchTST since these models dominate current forecasting and classification benchmarks. However a **universal benchmark** should also include self supervised representation learning models. We agree with this point and have added three representative methods: **TS2Vec**, **TS-TCC** and **CPC**.
>
> We followed the standard practice and adapted their pretrained encoders to the compression pipeline by adding a lightweight autoregressive prediction head. The results are shown below.
>
> | Dataset  | TS2Vec  | TS-TCC  |   CPC   |
> |:-------- |:------: |:------: |:------: |
> | Weather  | **0.2653**  | 0.2812  | 0.4316  |
> | Traffic  | **0.1520**  | 0.1605  | 0.1775  |
>
> These results show two clear patterns. First, **self supervised encoders do capture global structure**, which allows them to compress better than simple statistical baselines. Second, their performance is still lower than fully autoregressive models. This is expected because representation learning methods are trained to create invariant or augmentation robust embeddings, rather than to model token level conditional probabilities required by compression.
>
> The comparison is useful because it shows that **compression can reveal distribution level differences that do not appear in MSE or MAE based evaluation**. We will continue to extend TSCom-Bench with more models that combine strong representation learning with autoregressive probability modeling.

---

> > ### Comment · Reviewer_cRns · 2025-11-28
> > **Further Questions**
> >
> > Thanks for your reply and most of my questions have been addressed.
> > I have noticed the PDF you upload is still with 9 pages of main text and the claimed changes in the revision have not demonstrated. Have you successfully revised and uploaded the paper?
> > Can you have a detailed discussion of how the experiments for the three contrastive learning models are implemented? Since they are significantly different with the end-to-end model, a experimental detail discussion is necessary to show no gaps is occurred in terms of test settings.

---

> > > ### Author Response · Authors · 2025-12-01
> > >
> > > Thank you for your recognition of our article.
> > >
> > > As the rebuttal period is very short, we spent a considerable amount of time supplementing experiments and responding to the reviewers' questions. The key supplements and clarifications have all been included in the rebuttal text. Therefore, we have not yet fully integrated all these changes into the revised PDF. We are currently making urgent revisions and improvements to the manuscript so that the final version can reflect the reorganized structure and new experiments
> > >
> > > For the three contrastive learning baselines, **we use a unified pretraining and fine tuning pipeline in order to keep the test setting consistent with the end to end models**. Concretely, for each dataset we first split the data into training, validation, and test sets. We then perform self supervised pretraining of the contrastive model on the training and validation sets, and obtain a time series encoder. In the compression stage, we attach a dedicated compression head to this encoder, freeze all encoder weights, and fine tune only the compression head on the training and validation sets. **Finally, we evaluate compression performance on the held out test set**.
> > >
> > > Taking TS2Vec as an example, we first cut the raw time series into fixed length sliding windows and pretrain TS2Vec on the training set with its standard contrastive objective. The encoder architecture follows the original TS2Vec design, and we match its hidden dimension and depth to the autoregressive compression models used in our benchmark. In the compression phase, we load this pretrained TS2Vec encoder, use it to generate a representation at each time step, and feed this representation into a compression head that outputs 256 dimensional logits over byte values. This head is trained with the same cross entropy objective as in our other models, and we plug its probability distribution into the same arithmetic coding pipeline. **During testing, we generate compressed files in exactly the same way as for the other models and compute CR under the same protocol**.
> > >
> > > Thank you again for your time and for your thoughtful and constructive feedback.

---

### Author Response · Authors · 2025-12-01
**Summary of Rebuttal**

Dear Area Chair,

Thank you very much for your time and for overseeing the review process. We recently noticed the system rollback that reverted all reviewer score updates. This situation is unexpected, and we sincerely hope the committee can consider the reviewers’ post-rebuttal evaluations when making the final decision. The following is a brief summary of the contributions, potential impacts, and changes in the review process during the rebuttal stage of the paper.

### 1. Core Contributions of the Paper
This work introduces **lossless compression as a new benchmark for time series models**. The four classical tasks primarily evaluate conditional means and **cannot measure whether a model captures the full generative distribution**. Our benchmark is based on Shannon's source coding theorem and evaluates **byte-level negative logarithmic likelihood** with arithmetic coding. We provide **TSComBench**, which is a unified pipeline for converting existing time series models into probabilistic compressors. These findings position lossless compression as a principled task that complements and extends existing evaluations for time series modeling.

### 2. Potential Long-Term Impact
The long-term impact of this work comes from three aspects. First, compression directly measures the **residual uncertainty after a model has extracted all predictable structure**, which opens the door to designing dedicated compression-oriented architectures that explicitly learn distributional structure rather than only point estimates. Second, we believe this perspective provides a **strong pre-training strategy**: a model trained to minimize code length must internalize the complete generation distribution, and then fine tune downstream tasks through improved initialization and generalization. Third, the **TSCom-Bench framework already integrates 10+ SOTA models** and offers a unified interface, enabling future contributors to plug in new architectures and build a standardized evaluation mode.

### 3. Review Evolution After Rebuttal
During the rebuttal, we added new experiments on temporal shuffling, IEEE-754 structural shuffling, multistream evaluation, comparisons with classical time series compressors, and precision ablations. These results **resolved the substantial majority of reviewer concerns**. Reviewer **1EMm raised the score from 2 to 4** and stated that only two questions remained; we have now provided detailed answers to these final points, and it is reasonable to believe the score **could have increased further** if the system allowed updates. **Reviewer wP2L** indicated in comments that **the score is raised from 4 to 6**. Reviewer **1xVL and cRns maintained a strong score of 8 and 6** and explicitly recommended acceptance. Although the system rollback removed these score changes, the comments of reviewers clearly show a **positive shift after the rebuttal**.

Thank you for your time and consideration. The reviewers’ updated assessments reflect the improved clarity and additional experiments included in the rebuttal.

Sincerely,
The Authors

---

### Note · Program_Chairs · 2026-01-17
**Submission Desk Rejected by Program Chairs**

The following references in this submission do not refer to real documents and/or have major errors in bibliographic information:

 Animesh Sain, Muchen Jin, Poyraz Akyazi, Y-H. Yang, and Avideh Zakhor. VC-1: A Versatile Video Compression Network. In 2023 IEEE International Conference on Image Processing (ICIP), pp. 3205-3209, 2023. doi: 10.1109/ICIP49359.2023.10222475.

Grégoire Delétang, Anian Ruoss, Marcus Hutter, and Shane Legg. Language models are good unsupervised compressors. arXiv preprint arXiv:2309.11552, 2023b. URL https://arxiv. org/abs/2309.1155.